# Towards Better Branching Policies: Leveraging the Sequential Nature of Branch-and-Bound Tree

**Ce Zhang**[1,2], **Bin Zhang**[1,2] *, **Guoliang Fan**[1] *

[1] The Key Laboratory of Cognition and Decision Intelligence for Complex Systems,
Institute of Automation, Chinese Academy of Sciences
[2] School of Artificial Intelligence, University of Chinese Academy of Sciences
`{zhangce2023, zhangbin2020, guoliang.fan}@ia.ac.cn`

## Abstract

The branch-and-bound (B&B) method is a dominant exact algorithm for solving Mixed-Integer Linear Programming problems (MILPs). While recent deep learning approaches have shown promise in learning branching policies using instance-independent features, they often struggle to capture the sequential decision-making nature of B&B, particularly over long horizons with complex inter-step dependencies and intra-step variable interactions. To address these challenges, we propose Mamba-Branching, a novel learning-based branching policy that leverages the Mamba architecture for efficient long-sequence modeling, enabling effective capture of temporal dynamics across B&B steps. Additionally, we introduce a contrastive learning strategy to pre-train discriminative embeddings for candidate branching variables, significantly enhancing Mamba's performance. Experimental results demonstrate that Mamba-Branching outperforms all previous neural branching policies on real-world MILP instances and achieves superior computational efficiency compared to the advanced open-source solver SCIP. The source code can be accessed at `https://github.com/doctor-watson626/Mamba-Branching/`.

## 1 Introduction

Mixed Integer Linear Programming problems (MILPs) constitute a class of computationally challenging NP-hard problems with widespread applications across diverse domains, including scheduling (Chen, 2010), planning (Pochet & Wolsey, 2006), and transportation (Barnhart & Laporte, 2006). The branch-and-bound (B&B) method (Land & Doig, 2010) represents the predominant solution methodology for MILPs in practice. This approach begins with the relaxation of the original problem and iteratively branches on variables that violate integer constraints. By maintaining global upper and lower bounds, the method progressively converges toward an optimal solution. Many high-performance MILP solvers such as SCIP (Bolusani et al., 2024) and Gurobi (Gurobi Optimization, LLC, 2024) employ the B&B framework as their core solution architecture.

Within the B&B framework, the selection of branching variables plays a critical role in determining computational efficiency. To this end, learning-based branching methods have been proposed (Gasse et al., 2019; Gupta et al., 2020; 2022; Scavuzzo et al., 2022): by constructing a bipartite graph that incorporates instance features and intra-tree dynamic features, and utilizing graph convolutional networks (GCNN) (Gasse et al., 2019) for state encoding. Nevertheless, reliance on instance-specific features restricts their generalization to heterogeneous MILP instances. To enable cross-instance adaptability, recent approaches have focused on parameterizing the B&B tree to construct a shared feature space independent of specific problem data. For example, Zarpellon et al. (Zarpellon et al., 2021) develop a parameterized B&B tree framework to create a shared feature space, decoupling branching decisions from instance-specific features. Further advancing this approach, T-BranT (Lin et al., 2022) evaluates the mutual connections between candidate variables by the self-attention

---
*Corresponding to zhangbin2020@ia.ac.cn and guoliang.fan@ia.ac.cn

mechanism and employs Graph Attention Networks to encode the empirical branching history in the search tree.

However, existing works universally overlook the sequential nature inherent in B&B tree expansion. In this paper, our key insight lies in that the "branching path" from the root node to the optimal solution node essentially constitutes a serialization process. This "branching path", which encompasses the parameterized tree states and corresponding branching variables from each preceding step, significantly influences the current branching decision. While T-BranT incorporates historical data, it models the tree from an unordered graph perspective, failing to explicitly capture this essential sequential nature. Effectively modeling this sequential nature presents two key challenges: (1) Design of long sequence modeling architectures. The sequence model must simultaneously capture inter-step dependencies and intra-step candidate variable relationships. Given that each state comprises multiple candidate variables, the length of the sequence input will increase exponentially with the number of branching steps. Therefore, it is essential to develop specialized architectures that can accommodate ultra-long sequences. (2) Construction of discriminative feature embeddings. An embedding layer needs to be designed to map the features of candidate variables into a high-dimensional vector space with high discriminative power. This will enable the sequence model to effectively discern the dynamic evolution patterns of different variables.

To address these challenges, we propose Mamba-Branching. Mamba (Gu & Dao, 2024; Dao & Gu, 2024) is a novel network architecture characterized by its computational complexity that scales linearly with sequence length. This represents a significant improvement over the quadratic complexity associated with Transformers (Vaswani et al., 2017), making Mamba particularly well-suited for addressing challenge (1). Meanwhile, we employ contrastive learning to train the embedding layer prior to the overall imitation learning process, effectively tackling challenge (2). Experimental results demonstrate that Mamba-Branching outperforms all neural branching baselines across all real-world instances and achieves superior solving efficiency over the advanced open-source solver SCIP's default branching rule on challenging instances.

## 2 RELATED WORK

Learning-based approaches for accelerating MILP solving can be mainly divided into two paradigms (Bengio et al., 2021; Scavuzzo et al., 2024): replacing heuristic rules with neural networks within exact solution frameworks and employing neural networks as primal heuristics. Research under the first paradigm includes addressing branching variable selection (Khalil et al., 2016; Gasse et al., 2019; Gupta et al., 2020; 2022; Scavuzzo et al., 2022; Kuang et al., 2024a;b) and node selection (He et al., 2014; Labassi et al., 2022; Zhang et al., 2025) problems within the B&B framework, as well as tackling cut selection issues in cutting-plane algorithms (Tang et al., 2020; Huang et al., 2022; Wang et al., 2023). These methods solely employ neural networks to replace heuristic rules within exact solution frameworks, without compromising solution exactness. The second paradigm aims to efficiently produce high-quality feasible solutions—rather than exact solutions—to tighten the primal bound early in the process. A high-quality primal bound enables the B&B to eliminate a significant number of non-promising nodes at an early stage through its pruning process. This typically involves two key aspects: solution prediction (Ding et al., 2020; Nair et al., 2021; Khalil et al., 2022; Han et al., 2023; Huang et al., 2024; Liu et al., 2025) and neighborhood selection (Wu et al., 2021; Sonnerat et al., 2022; Huang et al., 2023; Yuan et al., 2025). The solution prediction approach typically employs neural networks to directly predict optimal solutions, then uses these predictions to guide the search process. Neighborhood selection starts from a feasible solution and fixes a subset of integer variables while optimizing the remainder, with neural networks selecting which variables to fix.

Our work focuses on the generalization of neural branching variable selection policies, particularly their ability to handle heterogeneous MILPs different from training instances. These approaches can be mainly divided into two categories: parameterizing the B&B tree and diversifying training instances. The first category aims to learn branching policies within a shared feature space across different MILP instances. TreeGate (Zarpellon et al., 2021) processes instance-independent features through a specialized neural architecture designed for branching decisions. Building on this, T-BranT (Lin et al., 2022) retains historical data, modeling it as a graph structure processed by Graph Attention Networks for current decision-making. The second category focuses on generating diverse

instances and incorporating them into the training of branching policies to enhance their generalization. AdaSolver (Liu et al., 2023) introduces adversarial instance augmentation, which generates more diverse instances in directions that hinder policy training. Meanwhile, MILP-Evolve (Li et al., 2025) proposes a novel LLM-based evolutionary framework capable of generating a set of diverse MILP classes with an unlimited number of instances. Specifically, our method falls into the first category.

## 3 PRELIMINARIES

### 3.1 B&B ALGORITHM AND BRANCHING RULES

**B&B Algorithm.** The standard form of MILPs is: $\arg\min_{\mathbf{x}} \left\{ \mathbf{c}^{\top}\mathbf{x} \mid \mathbf{A}\mathbf{x} \leq \mathbf{b}, \mathbf{x} \in \mathbb{Z}^{p} \times \mathbb{R}^{n-p} \right\}$, where the vector $\mathbf{x}$ represents $n$ variables to be optimized, with $p$ being the number of integer variables. $\mathbf{A}, \mathbf{b}, \mathbf{c}$ represent constraint matrix, constraint right term, and objective coefficient. For MILPs, an exact solution framework commonly used is B&B. This method first ignores the integer constraints to obtain and solve the relaxed problem at the root node. Subsequently, it iteratively searches for the global optimal solution through branching, bounding, and pruning. Branching involves selecting a variable with a fractional solution $x_j = b_j$ at the current node and adding the constraints $x_j \leq [b_j]$ and $x_j \geq [b_j] + 1$ to form two child nodes. During bounding, the global upper and lower bounds (also known as the primal and dual bounds) are determined based on all existing nodes. The pruning process eliminates obviously infeasible nodes according to these bounds. This procedure repeats until the upper and lower bounds converge, yielding the global optimal solution.

**Branching Rules.** Here, the branching variable selection during B&B significantly impacts solving efficiency by influencing tree size. In (Achterberg et al., 2005), several heuristic branching rules are introduced. Among them, strong branching evaluates candidate variables by creating child nodes for each candidate and selecting the one maximizing dual bound improvement. While highly effective, this approach incurs significant computational overhead that counteracts its benefits for solution speed. Pscost branching guides current branching by leveraging historical branching records, avoiding extra computation but performing poorly early in the search tree due to insufficient branching records. Hybrid approaches combine both methods' advantages by using strong branching initially to establish reliable branching patterns, then transitioning to pscost branching once sufficient historical data is accumulated. The state-of-the-art (SOTA) relpscost branching (Achterberg, 2025), SCIP's default rule, implements this strategy—a variable's pscost is only considered trustworthy after undergoing sufficient strong branching steps.

### 3.2 PARAMETERIZED B&B TREE

To parameterize the B&B tree and obtain instance-independent features for heterogeneous MILP problems, Zarpellon et al. (Zarpellon et al., 2021) design a state representation $s_t = (C_t, Tree_t)$. Here, $C_t \in \mathbb{R}^{|\mathcal{C}_t| \times 25}$ denotes candidate variable features, and $Tree_t \in \mathbb{R}^{61}$ represents tree features, where $\mathcal{C}_t$ denotes candidate variable set and $|\mathcal{C}_t|$ represents candidate variable number. Since all features reflect the dynamic process of B&B trees, all MILP instances can be processed uniformly in the same feature space by a neural network named TreeGate, which jointly processes candidate and tree features through two components: a candidate network and a tree network. The candidate network first embeds each variable's 25-dimensional features into an $h$-dimensional space, then progressively reduces the dimensionality from $h$ to $d$ through multiple layers that halve the dimension at each step. Meanwhile, the tree network projects the tree features $Tree_t$ into an $H$-dimensional space (where $H = h + h/2 + \ldots + d$) using a sigmoid activation to produce a gating vector $g \in [0,1]^H$. This gating vector modulates the candidate network's layer outputs through element-wise multiplication. The final output $e_t \in \mathbb{R}^{|\mathcal{C}_t| \times d}$ undergoes average pooling across the $d$-dimensional features, then being processed by a softmax layer to generate the candidate variable selection probabilities.

## 4 METHODOLOGY

In this section, we formally introduce Mamba-Branching, a neural branching policy specifically designed to capture the sequential structure of B&B trees. We begin by discussing the contrastive learning approach utilized for the embedding layer and the detailed design of the sequence inputs,

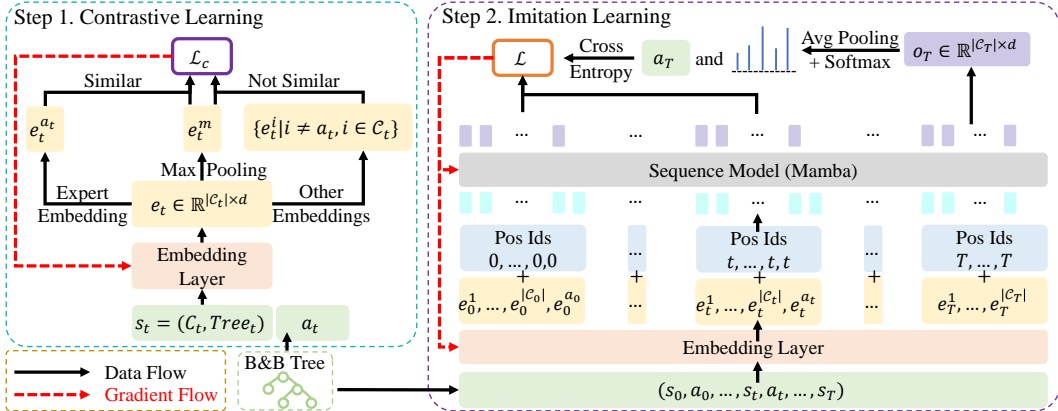

Figure 1: Overall framework of Mamba-Branching. The training process involves two stages: contrastive learning and autoregressive imitation learning. During the contrastive learning process, the state $s_t$ and expert decision $a_t$ at each branching step $t$ are used to train the embedding layer via the designed contrastive loss function $\mathcal{L}_c$. During imitation learning, the branching trajectory $(s_0, a_0, ..., s_T)$ is mapped to embeddings. At step $t$, expanded variable embeddings $(e_t^1, ..., e_t^{|\mathcal{C}_t|})$ and expert embedding $e_t^{a_t}$ form a group with shared positional encoding. These groups create a "branching path" input to the sequence model, where only outputs $o_t$ corresponding to the variable embeddings are selected, with $a_t$ serving as the label for imitation learning.

followed by the detailed implementation of imitation learning. The overall framework of Mamba-Branching is illustrated in Figure 1.

## 4.1 CONTRASTIVE LEARNING FOR EMBEDDING LAYER

The embedding layer serves as a critical interface between raw state representations and downstream sequence models. In natural language processing (NLP), the success of embedding techniques has been well-established. These methods leverage the inherent distinguishability of discrete word tokens, where each word's unique identity naturally translates to separable embedding vectors through standard training paradigms. (Bengio et al., 2000; Mikolov et al., 2013) However, the branching variable selection problem in B&B presents a fundamentally different challenge. The state representation at each branching step $t$, denoted as $s_t = (C_t, Tree_t)$ (see subsection 3.2), contains a set of candidate variables $\mathcal{C}_t$ that frequently exhibit remarkably similar feature characteristics. This high degree of intra-step similarity arises from the shared constraints and problem structure inherent in combinatorial optimization problems. Unlike the clear distinctions between words in NLP tasks, the subtle but decision-critical differences between candidate variables in B&B require a more sophisticated approach to embedding learning.

To address this challenge, we develop a principled framework for learning discriminative embeddings in B&B decision making. The core of our approach lies in recognizing that effective branching decisions require the embedding space to maintain consistent separation between selected and non-selected variables. We formalize this requirement through Proposition 1. This condition specifies that the similarity between the selected variable's embedding and a reference vector must exceed all other candidate similarities by a positive margin $\delta$.

**Proposition 1.** *For effective branching decisions, the embedding space must satisfy:*

$$\forall t, \exists \delta > 0 \ s.t. \ sim(e_t^a, e_t^m) \geq \max_{i \neq a} sim(e_t^i, e_t^m) + \delta, \tag{1}$$

*where $sim(\cdot)$ is a similarity judgment function, $e_t$ denotes embeddings, $e_t^m$ is an anchor, $a$ denotes the selected variable index.*

The intuitive understanding and proof of Proposition 1 are detailed in Appendix A. To approximately satisfy this condition during training, we design a contrastive loss $\mathcal{L}_c$, with the specific form as follows:

$$\mathcal{L}_c(\gamma) = \frac{1}{T} \sum_t (-\frac{e_t^m \cdot e_t^a}{\|e_t^m\|\|e_t^a\|} + \frac{1}{|\mathcal{C}_t| - 1} \sum_{i \neq a} \frac{e_t^m \cdot e_t^i}{\|e_t^m\|\|e_t^i\|}). \tag{2}$$

Here, we use cosine similarity as $\text{sim}(\cdot)$. $\gamma$ denotes the parameters of TreeGate, $T$ denotes the total number of branching steps, $e_t = \text{TreeGate}_\gamma(s_t)$ and $e_t^m$ is a context-aware anchor embedding (obtained via max-pooling over candidate embeddings).

The intuition behind this loss function design is to make the selected branching variable the most prominent and distinctive among all candidate variables. The max-pooling operation extracts a salient global feature as an anchor. By increasing the cosine similarity between the anchor and the selected branching variable while decreasing the cosine similarity between the anchor and other candidate variables, the loss amplifies their differences and drives the feature of the selected branching variable toward the globally most salient direction.

## 4.2 Sequential Modeling Design

In B&B tree, nodes are progressively expanded until the upper and lower bounds converge. This process can be viewed as navigating through a complex maze to find a "branching path" from the root node to the optimal solution node. Traditional neural approaches to branching decisions have predominantly relied on the immediate state of the tree, neglecting the historical sequence of visited nodes and prior branching choices. This myopic perspective is fundamentally limiting, as it fails to leverage the rich sequential information inherent in the branching process. Just as an effective maze-solving strategy requires reasoning about the entire traversed path to avoid dead ends and redundant exploration, optimal branching decisions demand a holistic understanding of the search trajectory. This underscores the imperative for a paradigm shift toward path-aware sequential modeling.

To effectively model branching decisions, the sequence model must capture not only the sequential progression of states but also the intricate interrelationships among candidate variables within each state. Therefore, we explicitly encode the features of each candidate variable at all branching steps. We formally define the branching path $\mathbf{S}$ as a structured sequence of embeddings, where each state at step $t$ is decomposed into its constituent candidate variables along with the selected branching decision. Specifically, $\mathbf{S}$ is represented as:

$$\mathbf{S} = [\underbrace{e_0^1, \ldots, e_0^{|\mathcal{C}_0|}, e_0^{a_0}}_{|\mathcal{C}_0|+1}, \ldots, \underbrace{e_t^1, \ldots, e_t^{|\mathcal{C}_t|}, e_t^{a_t}}_{|\mathcal{C}_t|+1}, \ldots, \underbrace{e_T^1, \ldots, e_T^{|\mathcal{C}_T|}}_{|\mathcal{C}_T|}], \tag{3}$$

where $|\mathcal{C}_t|$ denotes the number of candidate variables at branching step $t$, $a_t$ represents the index of the selected branching variable, $e_t$ denotes the embedding feature, and $T$ is the maximum number of branching steps in the branching path. This formulation ensures that both the sequential dynamics and the variable-level interactions are preserved, enabling the model to leverage granular features for improved decision-making.

To ensure temporal coherence across branching steps, we employ positional encodings that assign identical positional indices to embeddings within the same step. The complete input representation $\mathbf{S}'$ is constructed by combining the branching path $\mathbf{S}$ with a learnable positional encoding matrix $\mathbf{E}_{pos}$:

$$pos = [\underbrace{0, \ldots, 0, 0}_{|\mathcal{C}_0|+1}, \ldots, \underbrace{T, \ldots, T}_{|\mathcal{C}_T|}], \quad \mathbf{S}' = \mathbf{E}_{pos} \oplus \mathbf{S}, \tag{4}$$

where $\oplus$ denotes element-wise addition, $\mathbf{E}_{pos} \in \mathbb{R}^{(\sum_{t=0}^{T} |\mathcal{C}_t| + T) \times d}$ represents the learnable positional encoding matrix obtained by mapping $pos$.

Subsequently, $\mathbf{S}'$ is fed into Mamba to obtain the output $\mathbf{O}_t$:

$$\mathbf{O}_t = \text{Mamba}(\mathbf{S}') = [o_0^1, \ldots, o_0^{|\mathcal{C}_0|}, o_0^{a_0}, \cdots, o_t^1, \ldots, o_t^{|\mathcal{C}_t|}, o_t^{a_t}, \cdots, o_T^1, \ldots, o_T^{|\mathcal{C}_T|}], \tag{5}$$

within each group, only the outputs corresponding to $|\mathcal{C}_t|$ variable positions are extracted, denoted as $o_t = (o_t^1, \ldots, o_t^{|\mathcal{C}_t|})$, which are then processed through average pooling and softmax to obtain the variable probability distribution.

It can be observed that for the branching path $\mathbf{S}$, the sequence model actually needs to process an input length of $\sum_{t=0}^{T} |\mathcal{C}_t| + T$. When either $T$ or $|\mathcal{C}_t|$ becomes large, the length of $\mathbf{S}$ increases substantially, presenting significant challenges to the sequence model's ability to handle long sequences. Therefore, in addition to employing the most commonly-used Transformer Decoder as our sequence model, we also utilize Mamba (Gu & Dao, 2024) (see Appendix B for architectural details). In contrast

to the Transformer's quadratic complexity, Mamba achieves linear complexity relative to sequence length, making it unequivocally better suited for such long-sequence application scenarios. This is particularly critical in our application, where computational speed is paramount. If the neural branching policy's inference complexity becomes excessively high and computationally prohibitive, it would fundamentally undermine our original objective of acceleration.

## 4.3 Imitation Learning under Autoregressive Paradigm

Following prior works (Zarpellon et al., 2021; Lin et al., 2022), we employ relpscost branching as the expert to collect demonstration datasets for imitation learning. Unlike (Gasse et al., 2019; Gupta et al., 2020), we do not employ strong branching as the expert, for reasons detailed in Appendix C. For dataset collection, each instance is solved using SCIP. We sequentially record every state in the instance's tree along with the corresponding relpscost-selected branching decisions, resulting in a complete trajectory denoted as $(s_0, a_0, s_1, a_1, \ldots)$. In dataset $\mathcal{D}$, each instance's trajectory is partitioned into fixed-length sub-trajectories for storage.

In Mamba-Branching, the branching policy is defined as $\pi_\theta$, which operates in an autoregressive paradigm. The joint loss function $\mathcal{L}(\theta, \gamma)$ of embedding layer and sequence model is as follows: $\mathcal{L}(\theta, \gamma) = -\frac{1}{|\mathcal{D}|} \sum_{\tau \in \mathcal{D}} \sum_{(s_t, a_t) \in \tau} \log \pi_\theta(a_t | \tau_{0:t})$, where $\tau$ denotes a trajectory in $\mathcal{D}$, $\tau_{0:t} = (s_0, a_0, \ldots, a_{t-1}, s_t)$, and $|\mathcal{D}|$ represents the total number of trajectories in $\mathcal{D}$. During inference, the predictions from previous branching steps serve as input for the current step, yielding the probability distribution $\pi_\theta(\cdot | \hat{\tau}_{0:t})$ over candidate variables, where $\hat{\tau}_{0:t} = (s_0, \hat{a}_0, \ldots, \hat{a}_{t-1}, s_t)$.

## 5 Experiments

### 5.1 Setup

#### 5.1.1 Benchmarks

**MILP dataset.** Benefiting from instance-independent feature inputs, our method can generalize to heterogeneous MILPs, thus requiring distinct training and test instances to be selected separately. Following the selection of instances from previous works (Zarpellon et al., 2021; Lin et al., 2022), we construct two MILP datasets of different scales using instances from MIPLIB (Gleixner et al., 2021) and CORAL (Lehigh University COR@L Lab, n.d.): a smaller-scale dataset (MILP-S) and a larger-scale dataset (MILP-L). The MILP-S is entirely derived from (Zarpellon et al., 2021), comprising 19 training instances and 8 test instances. MILP-L extends (Lin et al., 2022) with 25 training and 73 test instances. Using SCIP as reference, we categorize the test set into 57 "easy" instances (solved in 20 minutes) and 16 "difficult" instances (more than 20 minutes). The details of MILP-S and MILP-L are provided in Appendix D.

**Branching Dataset Collection.** Consistent with prior works (Zarpellon et al., 2021; Lin et al., 2022), we use random branching for the first $r$ steps to improve B&B exploration, then switch to relpscost branching for data collection. For each training instance, we set $r \in \{0, 1, 5, 10, 15\}$ and collect training data using seeds $\{0, 1, 2, 3\}$, reserving seed 4 solely for validation.

#### 5.1.2 Metrics

**Nodes and Fair Nodes.** The node count in the B&B tree is a key performance indicator for branching policies, as it directly affects solving time. However, as observed in (Gamrath & Schubert, 2018), this measure can be influenced by side effects of sophisticated branching rules like strong branching. To address this, we also adopt the fair node count (Gamrath & Schubert, 2018), which removes such biases and offers a more precise assessment of a branching policy's effectiveness.

**Primal-Dual Integral.** Under a 1-hour time constraint, it is impossible to find the optimal solution for difficult instances in MILP-L. In such cases, neither time nor node count can serve as metrics, and the primal-dual integral (PD integral) serves as a more appropriate evaluation criterion (Gasse et al., 2022). With a time limit $T_l$, the PD integral is expressed as $\int_{t=0}^{T_l} \mathbf{c}^\top \mathbf{x}_t^\star - \mathbf{y}_t^\star \, \mathrm{d}t$, where $\mathbf{y}_t^\star$ is the best dual bound at time $t$, $\mathbf{x}_t^\star$ is the best feasible solution at time $t$.

Table 1: The experimental results on MILP-S and MILP-L. `blue background` indicates the best results, **bold** font indicates the best results in neural policies, and $\star$ denotes reaching the time limit (1 hour), OOM stands for CUDA out of memory.

| Method | MILP-S | | | MILP-L Easy | | | MILP-L Difficult |
|---|---|---|---|---|---|---|---|
| | Nodes | Fair Nodes | Time | Nodes | Fair Nodes | Time | PD Integral |
| Mamba-Branching | **2068.83** | **2091.54** | **111.78** | **1819.32** | **2053.91** | **123.79** | **12319.55** |
| TreeGate | 2146.75 | 2180.04 | 113.85 | 2218.29 | 2534.09 | 145.44 | 14625.68 |
| T-BranT | 2668.62 | 2715.16 | 165.59 | 2009.77 | 2298.76 | 210.27 | 13538.47 |
| Transformer-Branching | 3078.56 | 3120.04 | 3153.66 | OOM | OOM | OOM | OOM |
| GCNN | 33713.63$^\star$ | 33713.63$^\star$ | 730.89$^\star$ | - | - | - | - |
| random | 66487.87$^\star$ | 66487.87$^\star$ | 1189.57$^\star$ | - | - | - | - |
| pscost | 7308.23 | 7308.23 | 209.30 | 3681.06 | 3681.06 | 183.36 | 17097.06 |
| relpscost | 730.21 | 1227.25 | 82.51 | 667.93 | 1455.49 | 68.43 | 12741.36 |

### 5.1.3 BASELINES AND SETTINGS

We evaluate our method against two categories of branching policies: neural-based models and classical heuristics. Neural baselines include: (1) GCNN (Gasse et al., 2019), the seminal GNN-based branching policy; (2) TreeGate (Zarpellon et al., 2021) and T-BranT (Lin et al., 2022), instance-independent architectures serving as primary comparators to Mamba-Branching; (3) Transformer-Branching, a sequence-modeling baseline using Transformer, designed to isolate and highlight Mamba's architectural advantages. The heuristic rules include random, pscost, and relpscost, where relpscost serves as the expert and constitutes the upper bound of branching performance, though neural policies may surpass it in efficiency under certain conditions.

In our evaluation, we replace SCIP solver's (v8.0.4) branching policy with our neural branching policy. To isolate the study of branching policies and eliminate interference from other solver components, we disable all primal heuristics and provide each test instance with a known optimal solution value as a cutoff. However, during branching data collection, we intentionally omit the cutoff to obtain longer branching sequences. For neural baselines, during training, Mamba-Branching processes sequences up to $T = 99$ steps—corresponding to significantly long input lengths (see subsection 4.2). Due to excessive GPU memory consumption, Transformer-Branching is trained with a truncated horizon of $T = 9$. For evaluation consistency, all models generate decisions autoregressively under a unified budget of $T = 24$. More details for baselines and implementation can be found in Appendix E and Appendix F.

## 5.2 BRANCHING PERFORMANCE

The experimental results in MILP-S and MILP-L can be found in Table 1, with the fair node results of all neural branching policies per instance in MILP-S shown in Figure 2. Each instance is evaluated with 5 random seeds and the results are presented as geometric means. Notably, T-BranT necessitates at least one set of historical data, prompting the use of relpscost at the root node. This precise branching decision at the root significantly influences overall performance. For consistency, all neural policies use relpscost at the root node. To evaluate the performance of purely neural branching, we also evaluate variants without relpscost initialization denoted with the suffix "-p" (e.g., TreeGate-p, Mamba-Branching-p), as

Table 2: The neural branching results without relpscost initialization on MILP-S.

| Method | Nodes | Fair Nodes | Time |
|---|---|---|---|
| Mamba-Branching-p | **2272.43** | **2272.43** | **147.96** |
| Transformer-Branching-p | 5138.15 | 5138.15 | 5237.15 |
| TreeGate-p | 3179.55 | 3179.55 | 163.39 |
| T-BranT-p | 3922.63 | 3922.63 | 257.23 |

reported in Table 2. It can be observed that Mamba-Branching is the best branching policy besides relpscost. First, Mamba-Branching significantly outperforms the three lower-bound references: GCNN, random, and pscost. Compared with several neural branching policy baselines, whether initialized with relpscost or purely neural-based, Mamba-Branching surpasses T-BranT, TreeGate, and Transformer-Branching, achieving a new SOTA for neural branching policies. As shown in Figure 2, Mamba-Branching achieves the best performance on 3 of the 8 test instances, surpassing other neural branching policies. This demonstrates that the advantage of Mamba-Branching over other neural branching policies is not attributable to any single outlier but rather constitutes an overall superiority. Additionally, in terms of single-step inference time, as shown in Figure 3, Mamba significantly outperforms Transformer, highlighting its advantage as a sequence model.

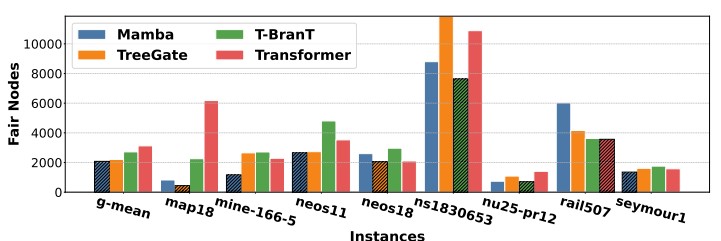 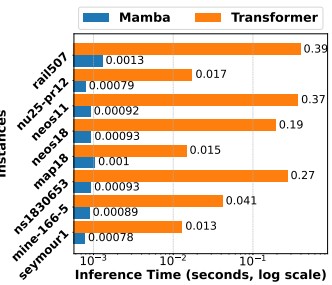

Figure 2: The fair node results of all neural branching policies in MILP-S. The black shading indicates the method with the fewest fair nodes for each instance.

Figure 3: The inference time comparison between Mamba and Transformer in MILP-S.

In MILP-L, we further evaluate several methods that demonstrated strong performance in MILP-S. For the 57 easy instances, performance is assessed using nodes, fair nodes and time. For the 16 difficult instances, we use the PD integral with a 1-hour time limit as the evaluation metric. The geometric mean results are presented in Table 1. The results demonstrate that for easy MILPs, Mamba-Branching remains the best neural branching policy, but still inferior to relpscost. For difficult instances, Mamba-Branching achieves the best PD integral performance among all branching policies, even surpassing relpscost. This indicates that within the same time limit, Mamba-Branching enables the fastest convergence of primal and dual bounds. Compared to MILP-S, Mamba-Branching demonstrates a more pronounced improvement over the previous best neural branching method (on solving time, 17% in MILP-L vs. 2% in MILP-S). Since MILP-L contains more test instances (57 vs. 8), the more substantial improvement by Mamba-Branching more clearly demonstrates its advantages from a statistical perspective. Furthermore, the violin plot in Figure 4 intuitively illustrates the distribution density of the solving time and presents the quantile information (e.g., for Mamba-Branching, 75% of the instances are solved in 390.8 seconds). Mamba-Branching achieves the lowest time at the 25th, 50th, and 75th percentiles, demonstrating superior and consistent performance across all difficulty levels. This indicates that its lower geometric mean results from robust, balanced improvements throughout the benchmark, not just a few outliers. More statistical analysis to show the advantages of Mamba-Branching is detailed in Appendix G. To further validate the performance of Mamba-Branching on more difficult instances, we have conducted a set of supplementary experiments—testing it on all difficult instances from the MIPLIB 2017 benchmark set that begin with the letter 'r'. Additional details are provided in Appendix I.

We also conduct a set of auxiliary verification experiments under a homogeneous scenario. Specifically, all training and testing instances are Set Covering, with only the parameters differing (Table 3). The results indicate that Mamba-Branching requires more nodes but achieves a shorter solving time compared to relpscost. It demonstrates that, in terms of solving efficiency, Mamba-Branching— which uses relpscost as an expert—has the potential to surpass it. Furthermore, since heterogeneous scenarios are inherently more challenging than homogeneous ones, Mamba-Branching's superiority over relpscost on difficult instances (despite being worse on easy ones) highlights its substantial promise and importance.

Table 3: Auxiliary verification results on Set Covering.

| Method | Mamba-Branching | TreeGate | relpscost |
|---|---|---|---|
| Nodes | **265.07** | 280.38 | 170.81 |
| Fair Nodes | **265.07** | 280.38 | 170.81 |
| Time | **9.64** | 9.89 | 11.96 |

### 5.3 DISCUSSION

**Advantage of Sequential Nature.** First, Mamba-Branching consistently outperforms TreeGate and T-BranT across all scenarios due to its consideration of the sequential nature of B&B trees. Neither TreeGate (which completely ignores historical data) nor T-BranT (which utilizes historical data non-sequentially) achieves the effectiveness of sequential historical data utilization. This aligns with our maze analogy in subsection 4.2: sequentially recalling paths facilitates better current decision-making.

**Limitation of Transformer.** Transformer-Branching also leverages sequential nature but performs poorly, with Transformer-Branching-p even underperforming the lower-bound pscost. The suboptimal performance stems from its 10-step branching history limit during training (due to hardware constraint), while Mamba-Branching accommodates 100 steps. Furthermore, the inference time

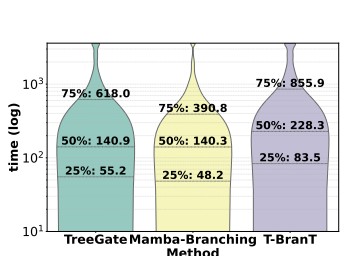

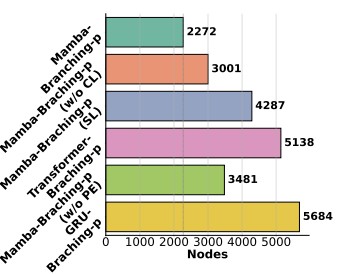

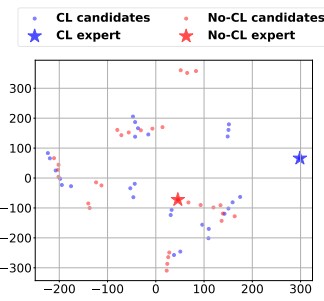

Figure 4: The violin plots for time on the easy instances in the MILP-L.

Figure 5: Ablation study results for the Mamba-Branching architecture on MILP-S.

Figure 6: At a random given state, the embeddings with and without contrastive learning.

comparison in Figure 3 demonstrates that in our time-sensitive scenario aimed at reducing solving time, Transformer is entirely unsuitable as a branching policy. The underlying reason here is that Transformer's complexity is quadratic with respect to sequence length, while Mamba's is linear. Although Transformer is theoretically suitable as a sequence model, employing Mamba offers greater practicality and feasibility. A more detailed comparison can be found in Appendix H.

**Factors Outperforming Relpscost.** As for relpscost, Mamba-Branching does not outperform in easy instances but surpasses it in difficult ones. The reason can be summarized as follows: (1) Relpscost is a hybrid method combining strong and pscost branching, incorporating a reliability criterion: a variable can only switch to pscost after being selected by strong branching a certain number of times. Therefore, for difficult instances with more variables, the initialization process is time-consuming, leading to potential inefficiency. In contrast, neural policies benefit from fast inference and exhibit advantages on difficult instances. (2) In relpscost, the use of pscost for leveraging historical data does not account for the sequential nature, whereas Mamba-Branching explicitly incorporates this consideration. (3) As mentioned in (Zarpellon et al., 2021), the relpscost in SCIP has been fine-tuned for a large number of instances, resulting in excellent performance on easy instances. However, for more complex and challenging instances, such parameter tuning may not provide adequate coverage.

## 5.4 ABLATION STUDY

In this section, an ablation study is conducted to quantify the impact of contrastive pretraining: (1) we compare Mamba-Branching-p with and without contrastive learning (Mamba-Branching-p w/o CL) to isolate CL's contribution to branching performance; (2) we introduce a supervised pretraining baseline (Mamba-Branching-p SL), in which embeddings are optimized via classification of expert-selected variables and the prediction head removed prior to downstream use. It serves as a discriminative representation learning alternative to CL; (3) we remove the positional encoding to highlight the effect of sequence modeling (Mamba-Branching-p w/o PE); and (4) in addition to the Transformer, we also employ a GRU as the sequence model (GRU-Branching). Results on MILP-S are reported in Figure 5. To validate CL's efficacy in enhancing embedding separability, we visualize learned representations via t-SNE (van der Maaten & Hinton, 2008) in Figure 6. We have also further investigated the impact of different forms of contrastive loss functions on the performance of Mamba-Branching, as detailed in Appendix J.

In terms of branching performance, Mamba-Branching-p significantly outperforms both its CL-ablated variant (w/o CL) and the supervised baseline (SL). While supervised learning offers a conceptually straightforward alternative, it proves insufficient for inducing embeddings that generalize effectively to the branching task. After removing the positional encoding of Mamba-Branching, the performance drops significantly, which underscores the importance of sequential modeling and aligns with the original motivation for proposing Mamba-Branching. Meanwhile, replacing the sequence model with either a GRU or a Transformer yields poor results, demonstrating the necessity of Mamba as the sequence model. Visualization via t-SNE further corroborates the necessity of contrastive pretraining. Under CL, embeddings of expert-selected variables exhibit greater separation and outlier-like distinctiveness relative to candidate variables. In its absence, these embeddings collapse toward the distribution of non-expert candidates, losing discriminative structure.

## 6 CONCLUSION AND FUTURE WORK

In this paper, we propose Mamba-Branching, the first approach to consider the sequential nature in B&B trees. To address the challenges of long sequences and embedding distinctiveness posed by sequential nature, we employ Mamba as the sequence model and design a contrastive learning method to train the embedding layer, enabling the sequence model to distinguish between different candidate variables. In experiments, Mamba-Branching outperforms all neural branching policies and achieves superior solving efficiency compared to relpscost on challenging instances. One limitation of our approach is the reliance on imitation learning, which requires a time-consuming collection of expert demonstrations. In future work, we will focus on investigating the potential of sequential nature in reinforcement learning-based branching policies, thereby eliminating the dependency on expert data.

## ETHICS STATEMENT

Our work does not involve any human subjects, animal experimentation, or the collection/use of personal data. We have reviewed the ICLR ethics guidelines and confirm that this submission raises no ethical concerns.

## REPRODUCIBILITY STATEMENT

We provide the following information, which is sufficient to support the reproducibility of Mamba-Branching. The specific methodology is detailed in section 4, and the proof of Proposition 1 is provided in Appendix A. In the experiments, the details of the benchmarks are described in subsubsection 5.1.1 and Appendix D, while hyperparameters and implementation details are provided in subsubsection 5.1.3 and Appendix F. The source code of Mamba-Branching and the datasets used can be found in `https://github.com/doctor-watson626/Mamba-Branching/`.

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

# A    Intuitive Understanding and Proof of Proposition 1

To further clarify the connection between Proposition 1 and contrastive learning approach, as well as the role of Proposition 1 in our methodology, we provide explanations from two perspectives: first, an intuitive explanation from the conceptual standpoint; second, an attempt to provide an explanation using mathematical formulations.

First, we make the following two assumptions, which will serve as the foundation for our subsequent analysis:

1. The feature embeddings of the optimal branch decision variable ($e_t^a$) can be distinguished from those of other branch variables ($e_t^i, \forall i \in \mathcal{C}_t, i \neq a$) in the vector space.

2. The feature embedding $e_t$ is informative for branch decision-making.

## A.1    Intuitive Explanation

Based on assumption 1, we establish an anchor point $e_t^m$ and employ contrastive learning to train the model such that: (1) the optimal branch variable's feature $e_t^a$ is pulled closer to $e_t^m$, while (2) other variables' features are pushed away from $e_t^m$. This approach amplifies the separation between optimal and non-optimal features, thereby ensuring sufficient distinguishability when these features are subsequently processed by the Mamba architecture.

## A.2    Mathematical Formulations

For each branching variable $i$, we define $F(i)$ as the branching score function (in this paper, the relpscost score) for variable $i$, such that the optimal branching variable satisfies

$$a = \arg\max_i F(i). \tag{6}$$

Assumption 2 posits that the embeddings can encode branching quality information, which we simplify to a linear relationship:

$$e_t^i = \alpha F(i) + \epsilon_i, \tag{7}$$

where $\alpha \in \mathbb{R}^d$ is a variable independent vector representing the quality-sensitive direction in the embedding space. $\epsilon_i \in \mathbb{R}^d$ is a noise term satisfying $\mathbb{E}(\epsilon_i) = 0$ with small magnitude $|\epsilon_i|$.

We use max pooling over the embeddings as the anchor point, defined as:

$$e_t^m = [..., \max_i e_t^i(j), ...] = [..., \max_i(F(i)\alpha(j) + \epsilon_i(j)), ...]. \tag{8}$$

For $e_t^a$, we have:

$$e_t^a = [..., \max_i F(i)\alpha(j) + \epsilon_a(j), ...]. \tag{9}$$

Indeed, if we assume that all $\epsilon_i$ terms are sufficiently similar in magnitude, we can derive the approximation $e_t^a \approx e_t^m$. Therefore, we can reasonably establish that

$$\forall i \neq a, \text{sim}(e_t^a, e_t^m) > \text{sim}(e_t^i, e_t^m). \tag{10}$$

This formulation is mathematically equivalent to the expression presented in Proposition 1:

$$\forall t, \exists \delta > 0 \text{ s.t. } \text{sim}(e_t^a, e_t^m) \geq \max_{i \neq a} \text{sim}(e_t^i, e_t^m) + \delta. \tag{11}$$

The proof effectively reflect the underlying intuition behind our design: to make the optimal variable's features closer to the anchor point while pushing other features farther away, thereby making the optimal variable's features more distinctive in the vector space.

# B  MAMBA ARCHITECTURE

Mamba is a novel network architecture based on State Space Model (SSM) that may potentially replace self-attention-based Transformer models (Gu & Dao, 2024; Dao & Gu, 2024; Wang et al., 2024). SSM is a concept originating from control theory, with its earliest roots traceable to the classical Kalman filter (Kalman, 1960). The continuous-time formulation of SSM can be represented as follows:

$$\dot{\mathbf{z}}(t) = \mathbf{M}(t)\mathbf{z}(t) + \mathbf{N}(t)\mathbf{u}(t)$$
$$\mathbf{y}(t) = \mathbf{P}(t)\mathbf{z}(t), \tag{12}$$

where $\mathbf{z}(t) \in \mathbb{R}^z$, $\mathbf{y}(t) \in \mathbb{R}^q$, $\mathbf{u}(t) \in \mathbb{R}^u$. After zero-order hold discretization, the discrete-time formulation is obtained as follows:

$$\mathbf{z}_t = \overline{\mathbf{M}}\mathbf{z}_{t-1} + \overline{\mathbf{N}}\mathbf{u}_t$$
$$\mathbf{y}_t = \mathbf{P}\mathbf{z}_t, \tag{13}$$

where $\overline{\mathbf{M}} = \exp(\Delta\mathbf{M})$, $\overline{\mathbf{N}} = (\Delta\mathbf{M})^{-1}(\exp(\Delta\mathbf{M}) - \mathbf{I}) \cdot \Delta\mathbf{N}$, $\Delta$ denotes the step size.

To meet the parallelization requirements of the training process, the SSM can alternatively be represented as follows:

$$\overline{\mathbf{K}} = (\mathbf{P}\overline{\mathbf{N}}, \mathbf{P}\overline{\mathbf{M}}\overline{\mathbf{N}}, \ldots, \mathbf{P}\overline{\mathbf{M}}^k\overline{\mathbf{N}})$$
$$\mathbf{y} = \overline{\mathbf{K}} * \mathbf{u}. \tag{14}$$

Mamba builds upon SSM by introducing Selective SSM, which essentially treats $\mathbf{N}$, $\mathbf{P}$ and $\Delta$ as functions of the input while keeping M unchanged. From a control theory perspective, this transforms the system from time-invariant to time-varying. Furthermore, Mamba incorporates hardware-aware algorithm design that enables efficient storage of intermediate results through parallel scanning, kernel fusion, and recalculation.

# C  REASONS FOR EMPLOYING RELPSCOST AS THE EXPERT

First, as mentioned in the original texts of (Zarpellon et al., 2021) and (Lin et al., 2022), strong branching is rarely applied in practical scenarios, and relpscost provides a more realistic expert representation. However, the rationale for selecting relpscost as the expert is not limited to this point alone. Here, we further analyze and present additional reasons necessitating the choice of relpscost as the expert.

The application scenario of our work differs fundamentally from that of GCNN: while GCNN focuses on training and testing on homogeneous instances, our method specifically targets testing on completely heterogeneous instances, emphasizing the generalization capability of the branching policy. Therefore, we emphasize that our approach fundamentally relies on the TreeGate neural network architecture and feature design specifically tailored for generalized branching policies. Under this framework, we experimentally validate the necessity of using relpscost as the expert policy and provide thorough analysis of the underlying rationale.

## C.1  COMPARATIVE EXPERIMENTS

Under TreeGate, we train policies using both strong branching and relpscost as experts, with results in Table 4. The results demonstrate that under the TreeGate framework, imitating strong branching achieves significantly lower accuracy compared to imitating relpscost. Meanwhile, on test instances, the policy imitating relpscost shows clearly superior performance.

## C.2  ANALYSIS AND CLARIFICATION

TreeGate emphasize incorporating tree-search-related features into the design without focusing on specific instances, while pseudo-costs align with the requirements of tree search by leveraging historical information. And relpscost essentially represents an improved form of pseudo-cost that incorporates historical strong branching experience.

Table 4: The imitation learning results using TreeGate as the network framework, with strong branching and relpscost serving as the experts respectively. The training was conducted for 50 epochs in all cases. The table reports the final network's loss and accuracy on both the training and test sets, as well as the solving time performance on test instances from (Zarpellon et al., 2021).

| Method | Train Loss | Test Loss | Train Accuracy | Test Accuracy | Solving Time |
|---|---|---|---|---|---|
| TreeGate + Relpscost | 0.9954 | 0.9907 | 0.7453 | 0.7561 | 137.41 |
| TreeGate + Strong Branching | 2.1845 | 2.5912 | 0.3590 | 0.2465 | 512.32 |

The features capture imperfect pseudo-costs while the relpscost labels contain refined pseudo-costs. The neural network learns to transform pseudo-costs from imperfect to refined versions. The feature design inherently results in better compatibility between TreeGate and relpscost, which consequently leads to the failure of TreeGate + strong branching.

Moreover, relpscost outperforms strong branching in terms of dataset collection speed. Although compared to the fundamental limitations of strong branching, this advantage might be relatively less significant.

## D  BENCHMARK DETAILS

All training and test instances in MILP-S are listed in Table 20. The training instances in MILP-L are presented in Table 21. All easy test instances in MILP-L are shown in Table 22, while all difficult test instances are presented in Table 23. Here, an instance is considered easy if SCIP's solving time is less than 20 minutes; otherwise, it is classified as difficult. These instances are sourced from MIPLIB (Gleixner et al., 2021) and CORAL (Lehigh University COR@L Lab, n.d.), all collected from real-world application scenarios, with specific instance selections referenced to (Zarpellon et al., 2021) and (Lin et al., 2022). Serving as benchmarks, these instances effectively reflect the practical significance of neural branching policies in real-world applications.

Actually, apart from the instances already present in (Zarpellon et al., 2021) and (Lin et al., 2022), we have only added 8 new challenging instances, all of which belong to the MILP-L. The criteria for selecting these eight challenging instances is as follows:

1. Instances marked as "hard" in the MIPLIB 2017 benchmark (Gleixner et al., 2021).

2. Instances with known optimal solutions and not infeasible, which can provide cutoff values.

3. A large number of nodes (>100) within the 1-hour time limit: If the number of nodes is too small, the time spent on solving the relaxed LP problem becomes excessively large. In such cases, the impact of branching rules may be negligible, making these instances unsuitable for comparing different branching policies.

4. The number of variables should be between 100 and 200k: For branching variable selection, a larger number of variables increases the difficulty of the selection. Therefore, the lower bound is set to 100 variables to sufficiently evaluate branching policies. However, if the number of variables is too large, all branching policies may perform poorly within the 1-hour time limit, making it impossible to distinguish their performance; hence, the upper bound is set to 200k.

Actually, in the MIPLIB 2017 benchmark set, there are only 19 instances labeled as "hard". After applying the aforementioned criteria for filtering, a total of 8 instances remain usable, and we have included all of them in MILP-L. Detailed information about these 19 hard instances—including the number of variables, objective value, and Nodes (within a 1-hour limit)—is provided in Table 5.

## E  DETAILED REASONS FOR BASELINE SELECTION

The neural branching policies and their selection rationale are detailed below: (1) GCNN (Gasse et al., 2019): This method is not designed for heterogeneous MILPs and performs poorly on unseen

Table 5: Information for all instances labeled as "hard" in the MIPLIB 2017 benchmark set. The details include: the objective value, the number of variables, the solving time and number of nodes using SCIP under a 1-hour time limit, and whether each instance was selected for MILP-L. For those not selected, the reason for exclusion is provided in the note.

| Instances Name | Objective | Variables | SCIP Time | Nodes | Note |
|---|---|---|---|---|---|
| bab2 | -357544.3115 | 147912 | 3600 | 251.6 | In MILP-L |
| bab6 | -284248.2307 | 114240 | 3600 | 502.6 | In MILP-L |
| cryptanalysisk b128n5obj14 | Infeasible | 48950 | | | Infeasible |
| highschool1 -aigio | 0 | 320404 | | | The number of variables exceeds 200k |
| markshare2 | 1 | 74 | | | The number of variables is fewer than 100 |
| neos-3402454 -bohle | Infeasible | 2904 | | | Infeasible |
| neos-3656078 -kumeu | -13172.2 | 14870 | 3600 | 57.6 | The number of nodes is fewer than 100 |
| neos-4338804 -snowy | 1471 | 1344 | 3600 | 201684.2 | In MILP-L |
| neos-4387871 -tavua | 33.38472993 | 4004 | 3600 | 6871.6 | In MILP-L |
| neos-4647030 -tutaki | 27265.706 | 12600 | 3600 | 7000.6 | In MILP-L |
| neos-5104907 -jarama | 935 | 345856 | | | The number of variables exceeds 200k |
| neos-5114902 -kasavu | 655 | 710164 | | | The number of variables exceeds 200k |
| nursesched -medium-hint03 | 115 | 34248 | 3600 | 483.4 | In MILP-L |
| opm2-z10-s4 | -33269 | 6250 | 3600 | 101.4 | In MILP-L |
| radiationm40 -10-02 | 155328 | 172013 | 3600 | 29963.6 | In MILP-L |
| s100 | -0.169723527 | 364417 | | | The number of variables exceeds 200k |
| supportcase10 | 7 | 14770 | 3600 | 1 | The number of nodes is fewer than 100 |
| supportcase19 | 12677206 | 1429098 | | | The number of variables exceeds 200k |
| thor50dday | 40417 | 106261 | 3600 | 1 | The number of nodes is fewer than 100 |

MILP instances outside the training distribution. The experimental results of GCNN highlight the advantage of instance-independent feature design in terms of generalization. (2) TreeGate (Zarpellon et al., 2021): The TreeGate network incorporates instance-independent inputs by design, making it suitable for heterogeneous MILPs. Additionally, since Mamba-Branching's embedding layer adopts the TreeGate architecture, TreeGate serves as a critical control group for our method. (3) T-BranT (Lin et al., 2022): Building upon TreeGate's feature design, T-BranT employs attention to capture mutual connections among candidates. Meanwhile, T-BranT processes historical data from an unordered graph perspective. The comparison with T-BranT serves to evaluate whether Mamba-Branching's sequential processing of historical data demonstrates superior performance over T-BranT's unordered graph approach. (4) Transformer-Branching: When selecting a sequence model for our approach, the Transformer would naturally be the most immediate consideration. Thus, we include Transformer-Branching as a comparative baseline against Mamba-Branching, specifically to highlight the advantages of employing Mamba as the sequence model.

The heuristic rules are selected with the following rationale: (1) Random: Serves as the performance lower bound, demonstrating the detrimental effects of completely omitting a deliberate branching policy. (2) Pscost: A purely historical data-driven branching method that, like Random, also establishes a performance lower bound. (3) Relpscost: The expert policy in the imitation learning of Mamba-Branching. Simultaneously, it is also the SOTA heuristic rule and the default rule in SCIP.

Table 6: The hyperparameters of Mamba-Branching

| Name | Description | Value |
|------|-------------|-------|
| d | Output dimension of the candidate net in TreeGate, which is equivalent to the embedding size of Mamba. | 8 |
| h | Hidden state dimension of the Candidate Net in TreeGate. | 64 |
| depth | Layer number of Tree Net in TreeGate. | 3 |
| batch_size | Batch size of Mamba Training | 32 |
| lr_cl | Learning rate of contrastive learning. | 0.0001 |
| optimizer_cl | Optimizer of contrastive learning. | Adam |
| lr | Learning rate of imitation learning. | 0.001 |
| optimizer | Optimizer of imitation learning. | AdamW |
| wd | Weight decay coefficient of imitation learning. | 0.01 |
| T_train | Maximum branching steps considered during training. | 99 |
| T_eva | Maximum branching steps considered during evaluating. | 24 |
| d_state | SSM state expansion factor in Mamba | 64 |
| d_conv | Local convolution width in Mamba | 4 |
| expand | Block expansion factor in Mamba | 2 |

Relpscost serves as the upper bound of decision accuracy for neural branching policies. However, benefiting from the fast inference speed of neural networks, neural branching policies may surpass relpscost in terms of efficiency.

## F  IMPLEMENTATION DETAILS

All experiments in this paper are run on NVIDIA A100-PCIE-40GB GPU and Intel(R) Xeon(R) Gold 5218 CPU. The hyperparameters in the training of Mamba-Branching are shown in Table 6.

During training, sequences consisting of every 100 branching steps are fed as input to Mamba. We systematically evaluate Mamba-Branching with varying branching path lengths during training, as shown in Table 7. The results reveal a clear performance peak at $T = 99$. Shorter sequences under-utilize the sequential information in B&B trees, while longer sequences exceed Mamba's effective processing capacity given our parameter constraints. Unlike standard Mamba implementations with large embeddings ($\geq 768$), we deliberately use a small embedding size (8) to prioritize inference speed—this design choice necessitates the optimal T=99 configuration.

During inference, the sequence fed into Mamba consists of: (1) states and predicted actions from the most recent 24 branching steps, and (2) the current state. We provide experimental results for step numbers of 5, 13, 25, 50, and 100, as shown in Table 8. As can be observed: 1) Sequence length grows substantially with path length, reaching 14,412 tokens at 100 steps—demonstrating why Mamba's linear complexity is essential for this task. 2) Node count exhibits a U-shaped curve: it initially decreases as longer contexts help the model better capture the sequential structure of the B&B tree, but then increases due to error accumulation under autoregressive inference, where predictions from previous steps are fed into subsequent steps. 3) The impact on solving time is more complex: it depends not only on the node count—which reduces LP computations—but also on the inference time, which rises as more real tokens are processed. Under the joint influence of these two factors, a branching path length of 25 emerges as the optimal choice, achieving the shortest total solving time.

## G  ADDITIONAL STATISTICAL RESULTS

Since the test instances we use are entirely different, and the solving time and node count vary significantly across instances, the geometric mean is adopted as a metric to mitigate the impact of excessively large or small values to some extent. However, to evaluate the advantage of Mamba-Branching over other baselines on each individual instance, rather than focusing solely on overall performance, more detailed statistical analysis is required. Therefore, we further conduct a fine-grained analysis of

Table 7: Results of training Mamba-Branching with varying Branching Path Lengths on MILP-S.

| Branching Path Length (T) | Nodes | Fair Nodes | Time | Actual Token Num (during training) | GPU Memory (during training) |
|---|---|---|---|---|---|
| 10 (9) | 2159.42 | 2188.49 | 118.09 | 2743.77 | 1.22 Gb |
| 100 (99) | 2068.83 | 2091.54 | 111.78 | 30667.39 | 7.94 Gb |
| 200 (199) | 3033.1 | 3085.48 | 164.28 | 63592.31 | 16.25 Gb |
| 500 (499) | 3575.88 | 3628.76 | 168.74 | 158533.33 | 35.89 Gb |

Table 8: In MILP-S, performance of Mamba-Branching and the actual number of tokens input to the Mamba model under different branching path lengths (number of steps: 5, 13, 25, 50, 100). For the metric 'real token num', we report the geometric mean over all instances of the maximum number of tokens required for each instance throughout the solving process.

| Branching Path Length | Nodes | Fair Nodes | Time | Actual Token Num |
|---|---|---|---|---|
| 5 | 2103.59 | 2126.81 | 111.97 | 986.1 |
| 13 | 2098.94 | 2120.28 | 113.05 | 2333.78 |
| 25 | 2068.83 | 2091.54 | 111.78 | 4132.02 |
| 50 | 2032.14 | 2054.39 | 112.77 | 7724.77 |
| 100 | 2091.46 | 2113.88 | 121.98 | 14412.15 |

Mamba-Branching on each instance through quartile tables, violin plots, win/tie/loss statistics tables, and significance testing.

### G.1 QUARTILE TABLES AND VIOLIN PLOTS

For neural branching policies, quartile tables and violin plots for time and nodes are shown in Table 9 Table 10 and Figure 7 Figure 8, respectively. They convey similar information: for example, Mamba-Branching achieves the time of 140.33 at the 50% in Table 9, meaning that 50% of the instances have solving time within 140.33. Additionally, violin plots intuitively illustrate the distribution density of the data across different values.

As shown, Mamba-Branching attains the smallest time and node count at the 25th, 50th, and 75th percentiles, indicating consistently superior performance across all difficulty ranges compared to the other two methods. This suggests that Mamba-Branching's lower geometric mean is not due to exceptional performance on a few extreme instances but rather stems from robust and balanced improvements across the entire benchmark.

### G.2 WIN/TIE/LOSS STATISTICS TABLE

In the win/tie/loss statistics table, we present the pairwise win/tie/loss outcomes between each method across all instances. For each instance, we consider the median value across five seeds. The results for the easy instances and difficult instances of MILP-L are shown in Table 11 Table 12 Table 13.

It can be observed that Mamba-Branching outperforms both TreeGate and T-BranT on both easy and hard instances, while also achieving superior performance over relpscost on difficult instances. This finding aligns with the original conclusions presented in the main text.

### G.3 SIGNIFICANCE TESTING

Here, we employ the Friedman test with Conover post-hoc analysis to perform significance testing. The Friedman test with Conover post-hoc analysis is a non-parametric statistical method for comparing multiple methods across multiple instances, especially when data may violate normality assumptions. Its procedure includes: (1) calculating instance-wise ranks; (2) conducting the Friedman test for overall significance, and if $p<0.05$; (3) applying Conover's test for detailed comparisons.

Table 9: For the easy instances in the MILP-L dataset (57 instances, each with 5 random seeds), the quartile table for time. For example, Mamba-Branching achieves the time of 140.33 at the 50%, meaning that 50% of the instances have solving time within 140.33. Meanwhile, the solving time contains difficulty information, and from this table, we can observe the adaptability of the method to problems of varying difficulties.

| Method | 25% | 50% | 75% |
|---|---|---|---|
| Mamba-Branching | 48.17 | 140.33 | 390.84 |
| T-BranT | 83.54 | 228.26 | 855.90 |
| TreeGate | 55.22 | 140.85 | 617.96 |

Table 10: For the easy instances in the MILP-L dataset (57 instances, each with 5 random seeds), the quartile table for Node counts.

| Method | 25% | 50% | 75% |
|---|---|---|---|
| Mamba-Branching | 536.0 | 2505.0 | 7827.0 |
| T-BranT | 645.0 | 3109.0 | 8122.0 |
| TreeGate | 911.0 | 2928.0 | 8069.0 |

Following the aforementioned procedure, we conduct overall significance tests on nodes, time, and pd integral, followed by post-hoc tests. The results are as follows:

1. Friedman test results for nodes (57 easy instances in MILP-L): Test statistic=2.502, p-value=0.2862, indicating no statistically significant differences.

2. Friedman test results for time (57 easy instances in MILP-L): Test statistic=18.000, p-value=0.0001, indicating statistically significant differences. Conover post-hoc analysis can be performed, as shown in Table 14.

3. Friedman test results for primal-dual integral (16 hard instances in MILP-L): Test statistic=8.625, p-value = 0.0347, indicating statistically significant differences. Conover post-hoc analysis can be performed, as shown Table 15.

On easy instances, in terms of runtime, Mamba-Branching demonstrates statistically significant differences compared to T-BranT (p=0.000014), while showing marginally significant differences with TreeGate (p=0.071287). For hard instances, Mamba-Branching has statistically significant differences vs. both TreeGate (p=0.011501) and T-BranT (p=0.007880), but no significant difference vs. relpscost (p=0.114340)—here, reference to earlier statistical results in Table 13 is needed. This is reasonable as its performance advantage over relpscost is smaller than over other methods.

## H    COMPUTATIONAL COMPLEXITY COMPARISON BETWEEN TRANSFORMER AND MAMBA

As is well-known, Mamba exhibits linear complexity with respect to sequence length, while Transformer demonstrates quadratic complexity. In this section, we present experimental results that provide a detailed comparison of the complexity between Mamba and Transformer when employed as branching policies. Our complexity analysis focuses on two key aspects: space complexity and time complexity. For space complexity, we compare the GPU memory consumption of Mamba and Transformer during both training and inference phases. Regarding time complexity, we primarily examine the inference latency of both models when functioning as branching policies. The experimental results are shown in Table 16.

As shown in the experiments, when processing 100 branching steps during training (even with a batch size of 1), Transformer-Branching fails to train altogether, while Mamba-Branching occupies minimal GPU memory. During inference, Mamba-Branching demonstrates significantly lower GPU memory consumption and inference time. In contrast, Transformer-Branching not only requires substantially more GPU memory but, more critically, suffers from prohibitively long inference times.

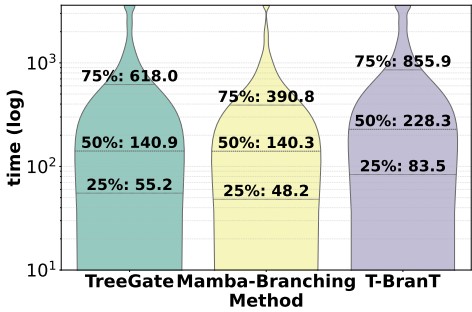 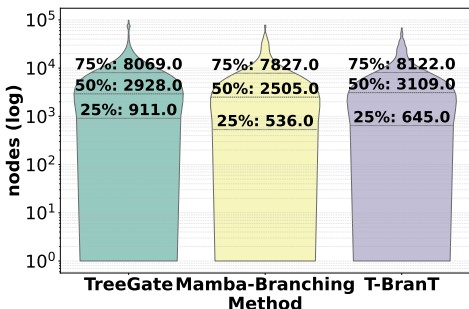

Figure 7: For the easy instances in the MILP-L dataset (57 instances, each with 5 random seeds), the violin plots for time.

Figure 8: For the easy instances in the MILP-L dataset (57 instances, each with 5 random seeds), the violin plots for nodes.

Table 11: For the easy instances in the MILP-L dataset (57 instances, each with 5 random seeds), the win/tie/loss table for node counts. Since 57 instances are too many to display their results directly, we use the win/tie/loss format to show the median performance (across 5 random seeds) of different methods on various instances when compared pairwise.

|  | TreeGate | Mamba-Branching | T-BranT |
|---|---|---|---|
| TreeGate | - | 24/1/32 | 30/2/25 |
| Mamba-Branching | 32/1/24 | - | 33/1/23 |
| T-BranT | 25/2/30 | 23/1/33 | - |

Since the fundamental purpose of adopting a neural branching policy is to accelerate MILP solving, such excessive inference time directly contradicts our original objective.

## I   MORE CHALLENGING INSTANCES

Here, we selected all instances from the MIPLIB 2017 benchmark set that begin with the letter 'r' and applied filtering criteria similar to our previous approach to identify suitable challenging instances for evaluating Mamba-Branching's performance.

We would like to emphasize that the selection of instances starting with the letter 'r' was solely based on their number in MIPLIB—evaluating too many instances would be prohibitively time-consuming, while too few would lack statistical significance. The distribution of instances is independent of their initial letters, which ensures the randomness of our selection. While we could have simply randomly sampled from all available instances, such an approach might raise concerns about "cherry-picking". In contrast, the set of instances starting with 'r' is explicitly listed at `https://miplib.zib.de/tag_benchmark.html`, making our selection process entirely transparent and verifiable, as we cannot omit any instances that might be unfavorable to our method.

The criteria for filtering challenging instances are as follows:

1. The solving time is greater than 20 minutes.

2. Instances must have a known optimal solution and must not be infeasible, ensuring that cutoff values can be provided.

3. Instances must generate a large number of nodes (>100) within the 1-hour time limit.

4. The number of variables should be between 100 and 200,000.

5. Not included in the training set.

For all instances beginning with the letter 'r', detailed information is provided in Table 17. As shown, a total of 9 instances meet our final criteria.

Table 12: For the easy instances in the MILP-L dataset (57 instances, each with 5 random seeds), the win/tie/loss table for time.

|  | TreeGate | Mamba-Branching | T-BranT |
|---|---|---|---|
| TreeGate | - | 23/0/34 | 37/0/20 |
| Mamba-Branching | 34/0/23 | - | 44/0/13 |
| T-BranT | 20/0/37 | 13/0/44 | - |

Table 13: For the hard instances in the MILP-L dataset (16 instances, each with 5 random seeds), the Win/Loss table for primal-dual integral.

|  | relpscost | Mamba-Branching | T-BranT | TreeGate |
|---|---|---|---|---|
| relpscost | - | 7/0/9 | 10/0/6 | 8/0/8 |
| Mamba-Branching | 9/0/7 | - | 14/0/2 | 13/0/3 |
| T-BranT | 6/0/10 | 2/0/14 | - | 10/0/6 |
| TreeGate | 8/0/8 | 3/0/13 | 6/0/10 | - |

The experimental results for solving these instances using Mamba-Branching and relpscost, respectively, are presented in Table 18. Results on the primal dual integral show that Mamba-Branching outperforms relpscost on a majority of instances (6 out of 9) and also achieves a better overall geometric mean. These results provide further evidence of the superior performance of Mamba-Branching across a broader set of instances.

## J    FURTHER DISCUSSION ON DIFFERENT CONTRASTIVE LOSSES

While we focus on the novel application of contrastive learning (CL) to branching variable selection, we acknowledge the importance of validating our specific CL design against standard alternatives. Therefore, we conducted additional experiments comparing our CL approach with two widely-used contrastive losses: InfoNCE and Triplet loss. Crucially, we kept identical definitions for anchors (max-pooled candidate embeddings), positive samples (expert-chosen variables), and negative samples (other candidates)—changing only the loss function formulation. Results are shown in Table 19.

These results demonstrate that our simple contrastive loss outperforms more sophisticated alternatives despite their theoretical advantages. We believe this validates our core insight: the primary innovation lies not in the loss function design but in how we define the contrastive structure for branching decisions. Our contribution is thus conceptual: we identify the appropriate elements to contrast in the B&B context and provide theoretical justification (Proposition 1) for this design. This principled approach to embedding discriminability, rather than loss function engineering, constitutes the true innovation of our contrastive learning module.

## THE USE OF LARGE LANGUAGE MODELS (LLMS)

In our paper, LLMs are utilized for writing, specifically for polishing the content to ensure standardized terminology and fluent writing.

Table 14: Conover post-hoc results for time (57 easy instances in MILP-L). The content of the table is p-values; if a p-value is <0.05, it can be considered that there is a significant difference.

|  | Mamba-Branching | T-BranT | TreeGate |
|---|---|---|---|
| Mamba-Branching | 1.000000 | 0.000014 | 0.071287 |
| T-BranT | 0.000014 | 1.000000 | 0.007329 |
| TreeGate | 0.071287 | 0.007329 | 1.000000 |

Table 15: Conover post-hoc results for primal-dual integral (16 hard instances in MILP-L).

|  | Mamba-Branching | T-BranT | TreeGate | relpscost |
|---|---|---|---|---|
| Mamba-Branching | 1.000000 | 0.011501 | 0.007880 | 0.114340 |
| T-BranT | 0.011501 | 1.000000 | 0.884271 | 0.310984 |
| TreeGate | 0.007880 | 0.884271 | 1.000000 | 0.247731 |
| relpscost | 0.114340 | 0.310984 | 0.247731 | 1.000000 |

Table 16: A comparison of computational complexity between Mamba-Branching and Transformer-Branching. During training, we uniformly set T_train=99 with a batch size of 1. For inference, we consistently use T_eva=24. After collecting 25 Branching steps, we measure the network's inference time and GPU memory consumption, take the geometric mean across all test instances of MILP-S.

| Method | GPU memory of Train (GB) | GPU memory of Inference (GB) | Inference Time (s) |
|---|---|---|---|
| Mamba-Branching | 0.017 | 0.013 | 0.00093 |
| Transformer-Branching | out of memory | 1.051 | 0.075 |

Table 17: For instances beginning with 'r', the information of objective, number of variables, number of nodes within the 1-hour time limit, and SCIP solving time. Each instance was evaluated against the aforementioned filtering criteria for identifying challenging instances, and the screening result is indicated in the "Note" column.

| Instances Name | Objective | Variables | SCIP Time | Nodes | Note |
|---|---|---|---|---|---|
| radiationm 18-12-05 | 17566 | 40623 | 2092.69s | 178498 | Satisfy |
| radiationm 40-10-02 | 155328 | 172013 | 3600 | 24578.8 | Satisfy |
| rail01 | -70.569964 | 117527 | 3600 | 24.6 | The number of nodes is fewer than 100 |
| rail02 | -200.44991 | 270869 | | | The number of variables exceeds 200k |
| rail507 | 174 | 63019 | 160.55 | 515 | The solving time of is fewer than 20min |
| ran14x18 -disj-8 | 3712 | 504 | 3108.4 | 184547.8 | Satisfy |
| rd-rplusc-21 | 165395.275 | 622 | 3600 | 113932.4 | Satisfy |
| reblock115 | -36800603 | 1150 | 3600 | 107606.4 | Satisfy |
| rmatr100-p10 | 423 | 7359 | | | Included in training set |
| rmatr200-p5 | 4521 | 37816 | | | Included in training set |
| rocI-4-11 | -6020203 | 6839 | 279.93 | 21304.8 | The solving time of is fewer than 20min |
| rocII-5-11 | -6.6755047 | 11523 | 3600 | 66194.4 | Satisfy |
| rococoB10 -011000 | 19449 | 4456 | 3600 | 15428.6 | Satisfy |
| rococoC10 -001000 | 11460 | 3117 | 3153.87 | 79096.4 | Satisfy |
| roi2alpha3n4 | -63.208495 | 6816 | 1432.63 | 11463.4 | Satisfy |
| roi5alpha10n8 | -52.322274 | 106150 | | | Included in training set |
| roll3000 | 12890 | 1166 | 44.27 | 1297.6 | The solving time of is fewer than 20min |

Table 18: Primal dual integral results of relpscost and Mamba-Branching on the filtered challenging instances beginning with 'r'. As can be observed, Mamba-Branching maintains its performance advantage over relpscost even on these difficult instances.

| Instances Name | relpscost | Mamba-Branching |
|---|---|---|
| radiationm18-12-05 | 143.17 | **128.27** |
| radiationm40-10-02 | 862.77 | **738.38** |
| ran14x18-disj-8 | **7082.15** | 9582.3 |
| rd-rplusc-21 | **359785.5** | 359786.96 |
| reblock115 | **2352.62** | 3914.59 |
| rocII-5-11 | 132899.86 | **129013.09** |
| rococoB10-011000 | 75874.98 | **34417.52** |
| rococoC10-001000 | 14484.75 | **2876.43** |
| roi2alpha3n4 | 23719.46 | **19886.59** |
| gmean of all | 10489.28 | **7371.76** |

Table 19: Mamba-Branching performance with different contrastive loss functions.

| Method | Nodes | Fair Nodes | Time |
|---|---|---|---|
| Mamba-Branching (ours) | 2068.83 | 2091.54 | 111.78 |
| + InfoNCE loss | 2182.88 | 2216.32 | 123.49 |
| + Triplet loss | 2322.41 | 2360.17 | 127.59 |

Table 20: All instances in MILP-S.

| Instance | Variables | Constraints | Set |
|---|---|---|---|
| air04 | 8904 | 823 | train |
| air05 | 7195 | 426 | train |
| dcmulti | 548 | 473 | train |
| eil33-2 | 4516 | 32 | train |
| istanbul-no-cutoff | 5282 | 20346 | train |
| l152lav | 1989 | 97 | train |
| lseu | 89 | 28 | train |
| misc03 | 160 | 96 | train |
| neos20 | 1165 | 2446 | train |
| neos21 | 614 | 1085 | train |
| neos-476283 | 11915 | 10015 | train |
| neos648910 | 814 | 1491 | train |
| pp08aCUTS | 240 | 246 | train |
| rmatr100-p10 | 7359 | 7260 | train |
| rmatr100-p5 | 8784 | 8685 | train |
| sp150x300d | 600 | 450 | train |
| stein27 | 27 | 118 | train |
| swath1 | 6805 | 884 | train |
| vpm2 | 378 | 234 | train |
| map18 | 164547 | 328818 | test |
| mine-166-5 | 830 | 8429 | test |
| neos11 | 1220 | 2706 | test |
| neos18 | 3312 | 11402 | test |
| ns1830653 | 1629 | 2932 | test |
| nu25-pr12 | 5868 | 2313 | test |
| rail507 | 63019 | 509 | test |
| seymour1 | 1372 | 4944 | test |

Table 21: Training instances in MILP-L

| Instance | Variables | Constraints |
|---|---|---|
| 30n20b8 | 18380 | 576 |
| air04 | 8904 | 823 |
| air05 | 7195 | 426 |
| cod105 | 1024 | 1024 |
| comp21-2idx | 10863 | 14038 |
| demulti | 548 | 290 |
| eil33–2 | 4516 | 32 |
| istanbul-no-cutoff | 5282 | 20346 |
| l152lav | 1989 | 97 |
| lseu | 89 | 28 |
| misc03 | 160 | 96 |
| neoS20 | 1165 | 2446 |
| neoS21 | 614 | 1085 |
| neos-476283 | 814 | 1491 |
| neos648910 | 11915 | 10015 |
| pp08aCUTS | 240 | 246 |
| rmatr100-p10 | 8784 | 8685 |
| rmatr100-p5 | 7359 | 7260 |
| rmatr200-p5 | 37816 | 37617 |
| roi5alpha10n8 | 106150 | 4665 |
| sp150 $\times$ 300d | 600 | 450 |
| stein27 | 27 | 118 |
| supportcase7 | 138844 | 6532 |
| swath1 | 6805 | 884 |
| vpm2 | 378 | 234 |

Table 22: Easy test instances in MILP-L , SCIP's solving time is less than 20 minutes.

| Instance | Variables | Constraints | SCIP Solving Time |
|---|---|---|---|
| aflow40b | 2728 | 1442 | 375.32 |
| app1-2 | 26871 | 53467 | 662.56 |
| bc1 | 1751 | 1913 | 237.68 |
| bell3a | 133 | 123 | 1.54 |
| bell5 | 104 | 91 | 0.63 |
| biella1 | 7328 | 1203 | 271.30 |
| binkar10_1 | 2298 | 1026 | 47.79 |
| blend2 | 353 | 274 | 0.37 |
| dano3_5 | 13873 | 3202 | 189.77 |
| fast0507 | 63009 | 507 | 150.11 |
| map10 | 164547 | 328818 | 515.00 |
| map18 | 164547 | 328818 | 250.49 |
| map20 | 164547 | 328818 | 218.78 |
| mik-250-20-75-4 | 270 | 195 | 55.67 |
| mine-166-5 | 830 | 8429 | 36.83 |
| misc07 | 260 | 212 | 28.94 |
| n2seq36q | 22480 | 2565 | 497.79 |
| neos11 | 1220 | 2706 | 171.35 |
| neos12 | 3983 | 8317 | 674.23 |
| neos-1200887 | 234 | 633 | 16.46 |
| neos-1215259 | 1601 | 1236 | 110.12 |
| neos13 | 1827 | 20852 | 96.20 |
| neos18 | 3312 | 11402 | 27.63 |
| neos-4722843-widden | 77723 | 113555 | 864.13 |
| neos-4738912-atrato | 6216 | 1947 | 304.32 |
| neos-480878 | 534 | 1321 | 52.92 |
| neos-504674 | 844 | 1344 | 114.11 |
| neos-504815 | 674 | 1067 | 34.61 |
| neos-512201 | 838 | 1337 | 42.39 |
| neos-584851 | 445 | 661 | 7.11 |
| neos-603073 | 1696 | 992 | 269.58 |
| neos-612125 | 9554 | 1795 | 43.31 |
| neos-612162 | 9893 | 1859 | 40.85 |
| neos-662469 | 18235 | 1085 | 566.6 |
| neos-686190 | 3660 | 3664 | 61.03 |
| neos-801834 | 3220 | 3300 | 51.71 |
| neos-803219 | 640 | 901 | 32.99 |
| neos-807639 | 1030 | 1541 | 20.52 |
| neos-820879 | 9522 | 361 | 56.56 |
| neos-829552 | 40971 | 5153 | 353.99 |
| neos-839859 | 1975 | 3251 | 64.10 |
| neos-892255 | 1800 | 2137 | 54.07 |
| neos-950242 | 5760 | 34224 | 149.60 |
| ns1208400 | 2883 | 4289 | 111.91 |
| ns1830653 | 1629 | 2932 | 148.27 |
| nu25-pr12 | 5868 | 2313 | 22.01 |
| nw04 | 87482 | 36 | 42.85 |
| p0201 | 201 | 133 | 0.81 |
| pg | 2700 | 125 | 45.39 |
| pp08a | 240 | 136 | 1.44 |
| rai507 | 63019 | 509 | 160.55 |
| roll3000 | 1166 | 2295 | 44.27 |
| rout | 556 | 291 | 39.51 |
| satellites1-25 | 9013 | 5996 | 952.04 |
| seymour1 | 1372 | 4944 | 61.91 |
| sp98ir | 1680 | 1531 | 92.98 |
| unitcal_7 | 25755 | 48939 | 889.85 |

Table 23: Difficult test instances in MILP-L , SCIP's solving time is more than 20 minutes.

| Instance | Variables | Constraints | SCIP Solving Time |
|---|---|---|---|
| atlanta-ip | 48738 | 21732 | 3600.00 |
| bab5 | 21600 | 4964 | 2665.41 |
| harp2 | 2993 | 112 | 1642.38 |
| map16715-04 | 164547 | 328818 | 2423.74 |
| msc98-ip | 21143 | 15850 | 3600.00 |
| mspp16 | 29280 | 561657 | 2722.23 |
| n3seq24 | 119856 | 6044 | 3600.00 |
| pigeon-10 | 490 | 931 | 3600.01 |
| bab2 | 147912 | 17245 | 3600.02 |
| bab6 | 114240 | 29904 | 3600.01 |
| neos-4338804-snowy | 1344 | 1701 | 3600.08 |
| neos-4387871-tavua | 4004 | 4554 | 3600.00 |
| neos-4647030-tutaki | 12600 | 8382 | 3600.09 |
| nursesched-medium-hint03 | 34248 | 14062 | 3600.00 |
| opm2-z10-s4 | 6250 | 160633 | 3600.01 |
| radiationm40-10-02 | 172013 | 173603 | 3600.00 |

