# OpenReview forum: "Towards Better Branching Policies: Leveraging the Sequential Nature of Branch-and-Bound Tree"
_ICLR.cc/2026/Conference — ICLR 2026 Poster_

### Official Review · Reviewer_usDM · 2025-10-28

**Soundness:** 3
**Presentation:** 3
**Contribution:** 3
**Rating:** 6
**Confidence:** 4

**Summary:**

### This paper presents ​Mamba-Branching, a novel neural branching policy for Mixed-Integer Linear Programming (MILP) that leverages the Mamba architecture to model the sequential nature of Branch-and-Bound (B&B) trees. The approach incorporates contrastive learning, which is a novel embedding layer trained with a contrastive loss to improve discriminability between candidate variables, which often have similar features. Experiments on heterogeneous MILP benchmarks (MILP-S and MILP-L) show that Mamba-Branching outperforms previous neural policies.

**Strengths:**

* This is the first work to explicitly model the sequential nature of B&B using a state-space model (Mamba). The idea of treating the search as a "branching path" is intuitive and well-motivated.
* The method is well-designed, combining contrastive learning for representation learning with sequential modeling for decision-making. The use of Mamba is justified given the long-sequence nature of the problem.
* The paper provides extensive experiments on real-world benchmarks, including easy and difficult instances. The results are interesting, demonstrating clear improvements over neural baselines.

**Weaknesses:**

1.  The description of the input sequence is confusing. It appears that at each step t, the model processes all candidate variable embeddings (e_t^1, ..., e_t^|C_t|) plus the expert's chosen variable embedding e_t^{a_t}. This leads to a sequence length of Σ(|C_t| + 1)over T steps.
    * Concern 1:​​ Why include the expert action e_t^{a_t}in the input during imitation learning? During inference, the expert action is unavailable. This creates a train-test mismatch. The model might be learning to rely on seeing the expert's choice in the history.
    * Concern 1:​​ The massive sequence length (Σ|C_t| + T) is acknowledged, but it's unclear how this is handled practically. How many tokens are typical for T=100? The paper mentions "tens of thousands," which seems prohibitive even for Mamba. Please provide statistics on average sequence lengths in training/inference.
2. The comparison to ​Transformer-Branching​ is unfair. Transformer-Branching is trained with only T = 9 steps due to hardware constraints, whereas Mamba-Branching uses T = 100. The performance gap is likely dominated by this difference in historical context rather than the architecture itself.
3. The rationale for choosing relpscostover strong branching as the expert is crucial but relegated to the appendix. The argument that TreeGate features are more compatible with relpscost is interesting, but needs a more straightforward explanation in the main text
4. The experimental setting on the difficult problem is limited. I would like to see more challenging instances being solved and compared to obtain a statistically persuasive experiment.

**Questions:**

1. During inference, does the input sequence contain the model's own previous predictions or the expert's actions? If it's the latter, how do you justify this setup for a practical scenario?
2. The results indicate that on standard ("easy") instances, all neural branching policies, including Mamba-Branching, still underperform the expert rule-based relpscost method. This raises two important questions: what explains this performance gap, and what would be needed for neural methods to become practically viable?
3. Why choose the difficult instances shown in Table 17? Is there justification for that, given the numerous difficult instances in MIPLIB 2017?

---

> ### Author Response · Authors · 2025-11-20
> **Summary of Advantages and Response to Weakness 1 (contains 2 Concerns)**
>
> We thank the reviewer for their positive and insightful assessment of our work. We appreciate their recognition of our novel approach in modeling B&B as a sequential decision-making process using a state-space model (Mamba), the well-designed methodology combining contrastive learning with sequential modeling, and the compelling experimental results demonstrated across diverse benchmark instances.
>
> Below, we provide a detailed explanation regarding the weaknesses and questions.
>
> > **Weakness 1 (Concern 1): The reason why expert actions are included in the input -- common practice in the autoregressive paradigm**
>
> In fact, this approach is a standard practice within the autoregressive paradigm in the field of Natural Language Processing (NLP). It has been widely and successfully applied in models like GPT and contemporary Large Language Models (LLMs).
>
> During training, ground-truth tokens provide context; during inference, the model uses its own predictions.
> This creates a known train-test discrepancy, but the remarkable success of LLMs validates this approach. Our core contribution is not solving this fundamental challenge but applying sequence modeling to branching decisions.
>
> > **Weakness 1 (Concern 2): Statistics on average sequence lengths in training/inference**
>
> First, we would like to point out that **Mamba is capable of handling long sequences**:
>
> - In the context of audio modeling tasks, **the original Mamba paper [1]** states : *“Figure 7 evaluates the effect of increasing training sequence lengths from 2^13 = 8192 to **2^20 ≈ 10^6**.”*
> - For DNA modeling tasks, **the original text [1]** notes: *“We pretrain models on sequence lengths 2^10 = 1024, 2^12 = 4096, 2^14 = 16384, 2^16 = 65536, 2^18 = 262144, **2^20 = 1048576**.”*
>
> Thus, it is evident that **Mamba possesses the ability to process extremely long sequences**.
>
> Subsequently, we provide information regarding the average sequence length (i.e., the actual number of tokens input to Mamba) during the training and testing (inference) of Mamba-Branching, as shown in Table 1.
>
> | Stage | Branching Path Length | Average Sequence Length (Actual Token Num)  |
> | :---- | :-------------------: | :-----------------: |
> | Train | 100 | 30745.65 |
> | Inference | 25 | 4132.02 |
>
> Table 1: The average sequence length (number of input tokens) for Mamba-Branching on MILP-S during training and inference. For the training phase, the value is computed by averaging the sequence length within each batch across all batches; for the inference phase, it is obtained by averaging over all test instances.
>
> Based on the above, the average sequence length during training for Mamba-Branching is **30,745.65**, while the maximum sequence length handled in the original Mamba paper is **1,048,576**. It is therefore **not surprising that Mamba performs well in our scenario**.
>
> Furthermore, in our response to **Reviewer 7jpx regarding "Weakness 3: Sequence length is chosen empirically,"** we conducted training with even longer branching path lengths. The results confirm that further increasing the sequence length indeed leads to performance degradation that Mamba cannot effectively handle, which is fully consistent with your viewpoint. This demonstrates that **our chosen setting (T=99) represents a validated performance peak** -- it **sufficiently leverages the sequential nature** of the branch-and-bound tree while **remaining within Mamba's effective** operating range before encountering the performance bottleneck associated with excessively long sequences.
>
> ### Reference
>
> [1] Gu, A., & Dao, T. (2024). Mamba: Linear-time sequence modeling with selective state spaces. In First Conference on Language Modeling. https://openreview.net/forum?id=tEYskw1VY2

---

> ### Author Response · Authors · 2025-11-20
> **Response to Weakness 2 and Weakness 3**
>
> > **Weakness 2: More comparison between Mamba-Branching and Transformer-Branching**
>
> As you pointed out, during training, the branching path lengths in Mamba-Branching and Transformer-Branching are different. Due to hardware constraints, the maximum length for Transformer-Branching is limited 10, which indicates significant limitations of the Transformer as a branching policy network. Meanwhile, the inference time of Transformer-Branching is excessively long, leading to a very long solving time on MILP-S (3153.66 seconds -- nearly 30 times that of Mamba-Branching), even though the number of nodes explored is acceptable. Given that the Transformer is clearly unsuitable as a branching policy network, we did not further investigate the performance comparison between the two models when both are trained with T=9.
>
> Certainly, we would be happy to supplement this set of experiments, comparing the results of Mamba-Branching and Transformer-Branching trained under T=9, as shown in Table 2.
>
> | Method               | Branching Nodes | Fair Nodes | Time    |
> |---------------------|-----------------|------------|---------|
> | Transformer         | 3078.56         | 3120.04    | 3153.66 |
> | Mamba-Branching     | 2159.42         | 2188.49    | 118.09  |
>
> Table 2: Results of Mamba-Branching and Transformer-Branching on MILP-S, under the condition of T=9 during training and T=24 during testing.
>
> It can be observed that, even when both are configured with T=9, Mamba-Branching still significantly outperforms Transformer-Branching, demonstrating the advantage of Mamba as a sequence model for branching variable selection.
>
>
>
> > **Weakness 3: Rationale for selecting relpscost as the expert is provided in the appendix**
>
> You are absolutely right that this part should be clearly explained in the main text. However, due to the page limit constraints of ICLR (9 pages), we had to place this entire section in the appendix. In the subsequent version, where ICLR allows the page limit to be extended to 10 pages, we will add a clear explanation for choosing relpscost as the expert in Section 4.3.

---

> ### Author Response · Authors · 2025-11-20
> **Response to Weakness 4: More challenging instances**
>
> > **Weaknesses 4: More challenging instances**
>
> First, in our response to Reviewer 7iig’s "Question 5: The specific selection logic for the instances used in the experiments", as well as in our reply to your Question 3, we have pointed out that: in fact, the MIPLIB 2017 benchmark set [2] (https://miplib.zib.de/tag_benchmark.html) contains only 19 instances labeled as "hard". After applying our filtering criteria, only 8 of these remain, and all of them have been included in MILP-L. Meanwhile, the remaining 8 challenging instances in MILP-L (making 16 in total) are entirely sourced from [1].
>
> However, we would be glad to provide additional experimental results on more challenging instances to further demonstrate the superiority of our method. Here, we selected all instances from the MIPLIB 2017 benchmark set that begin with the letter 'r' and applied filtering criteria similar to our previous approach to identify suitable challenging instances for evaluating Mamba-Branching's performance.
>
> We would like to emphasize that the selection of instances starting with the letter 'r' was solely based on their number in MIPLIB -- evaluating too many instances would be prohibitively time-consuming, while too few would lack statistical significance. The distribution of instances is independent of their initial letters, which ensures the randomness of our selection. While we could have simply randomly sampled from all available instances, such an approach might raise concerns about "cherry-picking". In contrast, the set of instances starting with 'r' is explicitly listed at https://miplib.zib.de/tag_benchmark.html, making our selection process entirely transparent and verifiable, as we cannot omit any instances that might be unfavorable to our method.
>
> The criteria for filtering challenging instances are as follows:
> - The solving time is greater than 20 minutes.
> - Instances must have a known optimal solution and must not be infeasible, ensuring that cutoff values can be provided.
> -Instances must generate a large number of nodes (>100) within the 1-hour time limit.
> - The number of variables should be between 100 and 200,000.
> - Not included in the training set.
>
> The specific rationale for these criteria is detailed in our response to Reviewer 7iig's Question 5: "The specific selection logic for the instances used in the experiments". For all instances beginning with the letter 'r', detailed information is provided in Table 3. As shown, a total of 9 instances meet our final criteria.
>
> The experimental results for solving these instances using Mamba-Branching and relpscost, respectively, are presented in Table 4. Results on the Primal Dual Integral show that Mamba-Branching outperforms relpscost on a majority of instances (6 out of 9) and also achieves a better overall geometric mean. These results provide further evidence of the superior performance of Mamba-Branching across a broader set of instances.
>
> ### Reference
> [1] Lin, Jiacheng, et al. "Learning to branch with Tree-aware Branching Transformers." Knowledge-Based Systems 252 (2022): 109455.
>
> [2] Gleixner, A., Hendel, G., Gamrath, G. et al. MIPLIB 2017: data-driven compilation of the 6th mixed-integer programming library. Math. Prog. Comp. 13, 443–490 (2021). https://doi.org/10.1007/s12532-020-00194-3

---

> ### Author Response · Authors · 2025-11-20
> **Supplementary Table 3 and Table 4 for weakness 4**
>
> | Instances Name       | Objective        | Variables | SCIP Time | Nodes     | Note                                |
> |---------------------|------------------|-----------|-------------------|-----------|-------------------------------------|
> | radiationm18-12-05  | 17566            | 40623     | 2092.69s          | 178498    | Satisfy                             |
> | radiationm40-10-02  | 155328           | 172013    | 3600              | 24578.8   | Satisfy                             |
> | rail01              | -70.5699643      | 117527    | 3600              | 24.6      | The number of nodes is fewer than 100 |
> | rail02              | -200.4499077     | 270869    |                   |           | The number of variables exceeds 200k |
> | rail507             | 174              | 63019     | 160.55            | 515       | The solving time of is fewer than 20min |
> | ran14x18-disj-8     | 3712             | 504       | 3108.4            | 184547.8  | Satisfy                             |
> | rd-rplusc-21        | 165395.2753      | 622       | 3600              | 113932.4  | Satisfy                             |
> | reblock115          | -36800603.23     | 1150      | 3600              | 107606.4  | Satisfy                             |
> | rmatr100-p10        | 423              | 7359      |                   |           | Included in training set            |
> | rmatr200-p5         | 4521             | 37816     |                   |           | Included in training set            |
> | rocI-4-11           | -6020203         | 6839      | 279.93            | 21304.8   | The solving time of is fewer than 20min |
> | rocII-5-11          | -6.675504732     | 11523     | 3600              | 66194.4   | Satisfy                             |
> | rococoB10-011000    | 19449            | 4456      | 3600              | 15428.6   | Satisfy                             |
> | rococoC10-001000    | 11460            | 3117      | 3153.87           | 79096.4   | Satisfy                             |
> | roi2alpha3n4        | -63.20849503     | 6816      | 1432.63           | 11463.4   | Satisfy                             |
> | roi5alpha10n8       | -52.32227435     | 106150    |                   |           | Included in training set            |
> | roll3000            | 12889.99999      | 1166      | 44.27             | 1297.6    | The solving time of is fewer than 20min |
>
> Table 3: For instances beginning with 'r', the information of objective, number of variables, number of nodes within the 1-hour time limit, and SCIP solving time. Each instance was evaluated against the aforementioned filtering criteria for identifying challenging instances, and the screening result is indicated in the "Note" column.
>
>
> | Instances Name      | relpscost   | Mamba-Branching |
> |---------------------|-------------|-----------------|
> | radiationm18-12-05  | 143.17      | **128.27**          |
> | radiationm40-10-02  | 862.77      | **738.38**          |
> | ran14x18-disj-8     | **7082.15**     | 9582.30         |
> | rd-rplusc-21        | **359785.50**   | 359786.96       |
> | reblock115          | **2352.62**     | 3914.59         |
> | rocII-5-11          | 132899.86   | **129013.09**       |
> | rococoB10-011000    | 75874.98    | **34417.52**        |
> | rococoC10-001000    | 14484.75    | **2876.43**         |
> | roi2alpha3n4        | 23719.46    | **19886.59**        |
> | gmean of all        | 10489.28    | **7371.76**         |
>
> Table 4: Primal dual integral results of relpscost and Mamba-Branching on the filtered challenging instances beginning with 'r'. As can be observed, Mamba-Branching maintains its performance advantage over relpscost even on these difficult instances.

---

> ### Author Response · Authors · 2025-11-20
> **Response to Question 1,2 and 3**
>
> > **Question 1: Model's own previous predictions or the expert's actions**
>
> Actually, besides performing a single relpscost branching (expert) at the root node, we subsequently adopt the model's own previous predictions. This ensures a fair comparison with T-BranT [1]—as they also perform one relpscost branching at the root node. And this method closely aligns with the core idea of strong branching/pseudo-cost hybrid branching (where relpscost is one of the most representative forms): applying strong branching near the root node and using pseudo-cost branching thereafter to leverage reliable early-stage historical data. Our approach, similarly, uses a single expert at the root node to initiate the subsequent input sequence, after which it relies entirely on the model's own previous predictions to form the input sequence.
>
> > **Question 2: What explains the performance gap, and what would be needed for neural methods to become practically viable?**
>
> **Performance gap explanation**
>
> The performance gap between neural branching policies and relpscost on standard ("easy") instances can be explained by understanding the fundamental differences between homogeneous and heterogeneous settings, as well as the computational characteristics of each approach.
>
> Key factors explaining the performance gap:
> - Setting differences: Our work focuses on the heterogeneous setting (different training and test instance structures), which is inherently more challenging than the homogeneous setting. Prior neural methods that outperform relpscost operate in homogeneous settings where training and test instances share similar structures.
> - Computational trade-offs: relpscost initializes with strong branching near the root node—a highly accurate but computationally expensive approach that evaluates each candidate variable by solving relaxation problems for child nodes. On simpler problems with fewer candidate variables, this overhead is minimal, giving relpscost an advantage.
> - Robustness to complexity: Mamba-Branching demonstrates superior robustness to problem difficulty. While relpscost's performance degrades significantly as problem complexity increases (due to expensive strong branching initialization), our method maintains consistent performance regardless of instance complexity.
>
> This explains our observed performance pattern: Mamba-Branching is slightly weaker than relpscost on simple instances but outperforms it on complex problems (as shown in our Primal-Dual Integral results on difficult MILP-L instances). Since real-world applications predominantly involve complex MILP instances, this robustness to complexity represents a critical advancement toward practical viability.
>
> **What would be needed for neural methods to become practically viable?**
>
> Neural branching methods are already practically viable in specific contexts, with the path forward differing by setting:
>
> Current practical applicability:
>
> - In homogeneous settings (similar problem structures), neural policies already outperform traditional methods like relpscost
> - In heterogeneous settings (our focus), Mamba-Branching represents a significant breakthrough by surpassing relpscost specifically on complex, real-world instances—precisely where computational efficiency matters most
>
> > **Question 3: Justification for the choice of instances**
>
> This selection process is transparent, reproducible, and aligned with prior work.
> In fact, this question is fully aligned with Question 5 raised by Reviewer 7iig: "The specific selection logic for the instances used in the experiments". Here, we briefly summarize our selection logic, with further details available in our response to Reviewer 7iig.
>
> Specifically, the MILP-S instances are taken entirely from [2], while MILP-L extends the instances from [1] by adding only 8 new challenging instances**. The selection criteria used in [2] and [1] can be found in the original publications and are also listed in our response to Reviewer 7iig.
>
> The filtering criteria for the 8 newly added challenging instances are as follows:
> - Instances marked as "hard" in the MIPLIB 2017 benchmark.
> - Instances with known optimal solutions and confirmed feasibility.
> - Instances requiring a large number of nodes (>100) within the 1-hour time limit.
> - Instances with variable counts between 100 and 200k.
>
> The rationale for these criteria is elaborated in our response to Reviewer 7iig. Notably, in the MIPLIB 2017 benchmark set (https://miplib.zib.de/tag_benchmark.html), only 19 instances are labeled as "hard". After applying the above filtering criteria, exactly 8 instances remain, all of which have been included in MILP-L.
>
> ### Reference
> [1] Lin, Jiacheng, et al. "Learning to branch with Tree-aware Branching Transformers." Knowledge-Based Systems 252 (2022): 109455.
>
> [2] Zarpellon, Giulia, et al. "Parameterizing branch-and-bound search trees to learn branching policies." Proceedings of the aaai conference on artificial intelligence. Vol. 35. No. 5. 2021.

---

> > ### Comment · Reviewer_usDM · 2025-11-27
> >
> > Thanks for the authors' clarification. I will keep my positive score.

---

> > > ### Author Response · Authors · 2025-11-27
> > >
> > > We sincerely appreciate your positive assessment of our work and thank you for the time and expertise invested in providing these valuable insights.

---

### Official Review · Reviewer_7iig · 2025-10-28

**Soundness:** 3
**Presentation:** 3
**Contribution:** 2
**Rating:** 4
**Confidence:** 4

**Summary:**

This paper proposes Mamba-Branching, a novel method for the Branch and Bound (B&B) algorithm in Mixed Integer Linear Programming (MILP) solvers. The authors' key insight is that existing neural branching methods (e.g., TreeGate and T-BranT) fail to effectively exploit the sequential nature of B&B tree expansion. To address this, they propose using the Mamba architecture (a selective state-space model) to model the complete branching path from the root to the current node. The method treats the historical decisions and tree states along the path as a long sequence. Experiments on the MILP-S and MILP-L datasets show that Mamba-Branching outperforms existing neural branching methods (TreeGate, T-BranT) in solving time, primal-dual integral (PD Integral), and number of nodes, and significantly surpasses Transformer-based models in computational efficiency.

**Strengths:**

+ Insightful problem formulation: The observation that traditional neural methods "generally overlook the sequential nature of B&B tree expansion" is a key insight. Modeling the "branching path" as a sequence is a novel and well-motivated idea that aligns with the intrinsic logic of the B&B process.
+ Outstanding computational efficiency: The paper clearly demonstrates the memory and speed advantages of Mamba-Branching in training and inference through Table 13. Compared to Transformer-Branching, it consumes significantly less memory and achieves an 80x faster inference speed, a remarkable advantage crucial for practical applications.

**Weaknesses:**

+ **Insufficient validation of the contrastive learning module**: The effectiveness of the contrastive learning (CL) module is inadequately validated. Although the paper introduces CL for embedding layer training in Section 4.1 and presents its effect via t-SNE visualization in Figure 6, the argument remains weak. The authors only compare with the "without CL" setting, which only proves that CL itself is helpful, but fails to demonstrate that their specific CL design is superior or necessary. To more comprehensively assess the value of their approach, the authors should compare their CL framework against several standard and widely-used contrastive learning loss functions and paradigms, such as InfoNCE.
+ **Ambiguous innovation attribution of Mamba**: The paper's demonstration of computational efficiency (Table 13, Figure 3) is excellent, strongly proving Mamba's significant advantage over Transformers in handling long sequential branching histories. However, these advantages primarily stem from the inherent properties of the Mamba architecture as an existing advanced model (e.g., linear complexity, selective state spaces), and the paper does not introduce any structural or algorithmic modifications or optimizations to the Mamba model itself. Therefore, the core innovation of the work leans more towards "successfully applying the cutting-edge sequence model Mamba to B&B."

**Questions:**

+ Are Mamba-Branching and baselines (e.g., TreeGate) compared using identical input features? It is recommended to disclose the feature list to ensure a fair comparison.
+ Has the method's generalization been validated on datasets beyond MILP-S/L, such as ML4CO 2021?
+ How does model performance vary with the length of the B&B path? Are there any observed performance bottlenecks for very long sequences?
+ Could the authors compare their approach with recent learning to branch methods [1, 2] to better position their contribution?
+ What is the specific selection logic for the instances used in the experiments?

**If the authors can convincingly demonstrate the method's generalization capability and the effectiveness of the contrastive learning component through additional experiments, I would be willing to update my score accordingly**.

[1] Zhang C, Ouyang W, Yuan H, et al. Towards imitation learning to branch for mip: A hybrid reinforcement learning based sample augmentation approach[C]// ICLR 2024.

[2] Parsonson C W F, Laterre A, Barrett T D. Reinforcement learning for branch-and-bound optimisation using retrospective trajectories[C]// AAAI 2023.

---

> ### Author Response · Authors · 2025-11-20
> **Summary of Advantages, Response to Weakness 1 and Weakness 2**
>
> We sincerely thank the reviewer for the thoughtful and constructive feedback. In particular, we appreciate the acknowledgment of our problem formulation-modeling the branching path in B&B as a sequence-which captures the intrinsic logic of the tree expansion process. Moreover, we are glad the reviewer highlighted the outstanding computational efficiency of Mamba-Branching, which achieves significantly lower memory consumption and an 80× faster inference speed compared to Transformer-based approaches, making it highly suitable for practical applications.
>
> Below, we address each concern in detail.
>
>
> > **Weakness 1: Insufficient validation of the contrastive learning module**
>
> We thank the reviewer for this valuable feedback. While our paper focuses on the novel application of contrastive learning (CL) to branching variable selection, we acknowledge the importance of validating our specific CL design against standard alternatives.
>
> In response, we conducted additional experiments comparing our CL approach with two widely-used contrastive losses: **InfoNCE** and **Triplet loss**. Crucially, we kept identical definitions for anchors (max-pooled candidate embeddings), positive samples (expert-chosen variables), and negative samples (other candidates)—changing only the loss function formulation.
>
> **Results on MILP-S:**
> | Method | Nodes | Fair Nodes | Time |
> |--------|-------|------------|------|
> | Mamba-Branching (ours) | **2068.83** | **2091.54** | **111.78** |
> | + InfoNCE loss | 2182.88 | 2216.32 | 123.49 |
> | + Triplet loss | 2322.41 | 2360.17 | 127.59 |
>
> These results demonstrate that our simple contrastive loss **outperforms more sophisticated alternatives** despite their theoretical advantages. We believe this validates our core insight: the primary innovation lies not in the loss function design but in **how we define the contrastive structure** for branching decisions.
>
> As shown in Figure 6 of our paper, our CL approach successfully creates distinctive embeddings for expert-selected variables compared to candidates, with the t-SNE visualization confirming improved separation. While InfoNCE and Triplet losses introduce additional hyperparameters (temperature coefficient and margin) that could potentially improve performance with extensive tuning, our approach achieves superior results **without such optimization**-demonstrating its practical efficacy and robustness.
>
> Our contribution is thus conceptual: we identify the appropriate elements to contrast in the B&B context and provide theoretical justification (Proposition 1) for this design. This principled approach to embedding discriminability, rather than loss function engineering, constitutes the true innovation of our contrastive learning module.
>
>
> > **Weakness 2: Ambiguous innovation attribution of Mamba**
>
> We appreciate the reviewer’s accurate summary—but clarify that applying Mamba is not our core innovation. Our key insight is recognizing the B&B process as a long sequential “branching path” and designing a policy that explicitly models this path.
>
> Our actual logic is as follows:
> 1. First, we highlight the importance of "branching paths" and argue that they should be incorporated into the design of the branching policy network.
> 2. Next, we propose that using a sequence model to handle these branching paths is highly appropriate.
> 3. Subsequently, we analyze that branching paths can result in long sequential inputs, and Mamba is theoretically suitable for processing such long sequences due to its linear complexity.
> 4. Finally, we compare the performance of GRU, Mamba, and Transformer as sequence modeling backbones, with experimental results demonstrating that Mamba is indeed the most suitable choice.
>
> The reviewer suggested that "the core innovation of the work leans more towards 'successfully applying the cutting-edge sequence model Mamba to B&B'". However, we would like to clarify that this is **not the central innovation** of our work. Our **core innovation** lies in highlighting the importance of branching paths and sequence modeling for the design of branching policies. We start from the intrinsic sequential nature of B&B and integrate this understanding into the network design. As for the **"successful application of Mamba,"** this is **a conclusion validated by our experimental results, rather than the central innovation itself**.
>
> Furthermore, our approach goes beyond a direct application of Mamba. To enhance Mamba's adaptability to the branching problem, we have also introduced specific designs:
> - Input Sequence Design: In the Mamba's input sequence, all candidate features at each time step are jointly fed into the network. We assign the same positional encoding to all features within the same time step to distinguish features across different time steps.
> - Model Size: To accelerate inference speed, we reduce the number of network parameters and set the embedding size to 8 (compared to ≥768 in the original paper).

---

> ### Author Response · Authors · 2025-11-20
> **Response to Questions 1 and Questions 2**
>
> > **Questions 1: Are Mamba-Branching and baselines (e.g., TreeGate) compared using identical input features? It is recommended to disclose the feature list to ensure a fair comparison.**
>
> Yes. As stated in Line 188 and Section 3.2, Mamba-Branching uses the exact same parameterized B&B tree representation as TreeGate. This ensures fair comparison with TreeGate/T-BranT, our primary baselines. Specifically, their feature design consists of the candidate state and the search tree state, with state dimensions of 25 per candidate variable and 61 for the search tree, respectively. These features are largely associated with the dynamics of the branch-and-bound tree. A detailed description of TreeGate's input features is provided in Appendix B of the original TreeGate paper [1].
> Since this feature design constitutes a core contribution of their work, rather than ours, we did not include a detailed description of the input features in our manuscript.
>
> > **Questions 2: Questions 2: Has the method's generalization been validated on datasets beyond MILP-S/L, such as ML4CO 2021?**
>
>
> We appreciate the reviewer's question about broader validation. However, evaluating Mamba-Branching on ML4CO 2021 would be fundamentally inappropriate due to a mismatch in problem settings:
>
>
> ML4CO is designed for homogeneous scenarios. To support this point, we directly quote from the ML4CO paper [3]:
>
> *"A problem benchmark consists of a collection of MILP instances in the standard MPS ﬁle format. Each benchmark was split into a training and a test set, originating from the same problem distribution."*
>
> In contrast, our method targets heterogeneous generalization across structurally different MILP instances. These settings require fundamentally different architectural approaches:
>
> - In homogeneous settings, methods like GCNN [2] incorporate instance-specific static features to achieve strong performance
> - In heterogeneous settings, architectures like TreeGate [1] and our Mamba-Branching deliberately exclude static features to enable cross-instance generalization
>
> While GCNN achieves excellent results on ML4CO (homogeneous), it fails catastrophically on heterogeneous instances as shown by its 33,713 nodes on MILP-S (Table 1 in original paper) - demonstrating why setting alignment matters.
>
> Nevertheless, we did validate generalization on more challenging instances.
> In response to Reviewer usDM's "Weaknesses 4: More challenging instances", we conducted tests on more difficult instances. Specifically, we selected challenging instances from the MIPLIB 2027 benchmark set that begin with the letter 'r'. The rationale for this selection approach is detailed in our response. In summary, it ensures full transparent and verifiable while providing a suitable number of instances. **On these instances, Mamba-Branching maintains its performance advantage over relpscost**.
>
> ### Reference
> [1] Zarpellon, Giulia, et al. "Parameterizing branch-and-bound search trees to learn branching policies." Proceedings of the aaai conference on artificial intelligence. Vol. 35. No. 5. 2021.
>
> [2] Gasse, Maxime, et al. "Exact combinatorial optimization with graph convolutional neural networks." _Advances in neural information processing systems_ 32 (2019).
>
> [3] Gasse, M., Bowly, S., Cappart, Q., Charfreitag, J., Charlin, L., Chételat, D., Chmiela, A., Dumouchelle, J., Gleixner, A., Kazachkov, A.M., Khalil, E., Lichocki, P., Lodi, A., Lubin, M., Maddison, C.J., Christopher, M., Papageorgiou, D.J., Parjadis, A., Pokutta, S., Prouvost, A., Scavuzzo, L., Zarpellon, G., Yang, L., Lai, S., Wang, A., Luo, X., Zhou, X., Huang, H., Shao, S., Zhu, Y., Zhang, D., Quan, T., Cao, Z., Xu, Y., Huang, Z., Zhou, S., Binbin, C., Minggui, H., Hao, H., Zhiyu, Z., Zhiwu, A. &amp; Kun, M.. (2022). The Machine Learning for Combinatorial Optimization Competition (ML4CO): Results and Insights. <i>Proceedings of the NeurIPS 2021 Competitions and Demonstrations Track</i>, in <i>Proceedings of Machine Learning Research</i> 176:220-231 Available from https://proceedings.mlr.press/v176/gasse22a.html.

---

> ### Author Response · Authors · 2025-11-20
> **Response to Questions 3 and Questions 4**
>
> > **Question 3: How does model performance vary with the length of the B&B path?**
>
> Thank you for this insightful question! Indeed, we treated the length of the branching path (i.e., the number of branching steps) as a hyperparameter and tuned it during our experiments, **ultimately selecting 25 steps (t = 0 to 24)**. However, as you rightly pointed out, it is also valuable to discuss the performance under different path lengths and the associated performance bottlenecks. Here, we provide experimental results for step numbers of 5, 13, 25, 50, and 100, as shown in Table 2.
>
>
>
> | path length | nodes   | fair nodes | time  | real token num |
> |----------|---------|------------|-------|----------------|
> | 5        | 2103.59 | 2126.81    | 111.97| 986.10         |
> | 13       | 2098.94 | 2120.28    | 113.05| 2333.78        |
> | 25       | 2068.83 | 2091.54    | 111.78| 4132.02        |
> | 50       | 2032.14 | 2054.39    | 112.77| 7724.77        |
> | 100      | 2091.46 | 2113.88    | 121.98| 14412.15       |
>
> Table 2: In MILP-S, performance of Mamba-Branching and the actual number of tokens input to the Mamba model under different branching path lengths (number of steps: 5, 13, 25, 50, 100). For the metric 'real token num', we report the geometric mean over all instances of the maximum number of tokens required for each instance throughout the solving process.
>
>
> As can be observed from Table 2:
> 1. Sequence length grows substantially with path length, reaching 14,412 tokens at 100 steps - demonstrating why Mamba's linear complexity is essential for this task.
>
> 2. Node count follows a U-shaped curve. The initial decrease occurs because a longer context allows the sequence model to better capture the sequential nature of the B&B tree. However, under the autoregressive paradigm, the model can only use its prediction from the previous step as the input for the next step during inference. This inevitably leads to error accumulation, which subsequently causes the number of nodes to increase again.
>
> 3. The impact on solving time is more complex: it is influenced not only by the number of nodes (which results in fewer LP computations) but also by the inference time (which increases with a larger real token num). Under the combined effect of these two factors, a branching path length of 25 proves to be the most suitable choice, yielding the shortest overall solving time.
>
> In conclusion, as the reviewer rightly pointed out: from the perspective of node count, indiscriminately increasing the branching path length introduces a performance bottleneck. Furthermore, solving time depends on a complex interplay between reducing the node count and minimizing inference time. Thus, a branching path length of 25 emerges as the optimal balance considering these competing factors.
>
> > **Question 4: Could the authors compare their approach with recent learning to branch methods [1, 2] to better position their contribution?**
>
> Regarding the two papers [1,2] the reviewer mentioned, we would like to clarify that **both employ GCNN as the neural network architecture and are evaluated under homogeneous settings** (as discussed in response to Question 2). Their features include instance-specific graph structure, which prevents generalization to unseen problem types (as shown by GCNN’s poor MILP-S/L performance in Table 1 of the original paper). To support this point, we directly quote from the benchmark selection sections of [1] and [2]:
>
> - In [1]: *"We perform evaluation on popular binary integer programming problems: set covering, combinatorial auctions, maximum independent set, and capacitated facility location."* Additionally, *"we conduct experiments on more difficult MIP problems from the ML4CO 2021 competition."*
>
> - In [2]: *"In total, we considered four NP-hard problem benchmarks: set covering, combinatorial auction, capacitated facility location, and maximum independent set."*
>
> Since our focus is heterogeneous generalization, direct comparison is inappropriate—different goals, architectures, and evaluation protocols.
>
> ### Reference
>
> [1] Zhang, Changwen, et al. "Towards imitation learning to branch for mip: A hybrid reinforcement learning based sample augmentation approach." _The Twelfth International Conference on Learning Representations_. 2024.
>
> [2] Parsonson, Christopher WF, Alexandre Laterre, and Thomas D. Barrett. "Reinforcement learning for branch-and-bound optimisation using retrospective trajectories." _Proceedings of the AAAI Conference on Artificial Intelligence_. Vol. 37. No. 4. 2023.

---

> > ### Author Response · Authors · 2025-11-23
> > **Supplementary experiments on the open-source work [2] for Question 4**
> >
> > > **Supplementary experiments on the open-source work [2]**
> >
> > In the two papers you mentioned, [1] is not open-source, while Retro-Branching [2] is open-source. We have also supplemented the experimental results of [2] on the MILP-S benchmark, as shown in the table 4.
> >
> > | Name        | Retro Time (s) | Retro Nodes | Mamba Time (s) | Mamba Nodes |
> > |-------------|-------------------|-------------|----------------|-------------|
> > | seymour1    | 3600.122          | 95395.0     | 52.996         | 1306        |
> > | mine-166-5  | 3600.059          | 182076.0    | 116.448        | 3465        |
> > | ns1830653   | 2635.406          | 154005.0    | 344.643        | 9009        |
> > | map18       | 1174.547          | 6741.0      | 197.823        | 641         |
> > | neos18      | 1006.507          | 74476.0     | 45.674         | 2840        |
> > | neos11      | 1168.115          | 31874.0     | 203.518        | 3075        |
> > | nu25-pr12   | 891.808           | 38292.0     | 15.888         | 493         |
> > | rail507     | 3600.413          | 29586.0     | 543.561        | 3521        |
> > |gmean	|1872.98	|51370.12	|116.43	|2112.19
> >
> > Table 4: Results of Mamba-Branching and Retro-Branching [2] on MILP-S (1-hour time limit). Due to time constraints, we executed each instance using only seed=0; nevertheless, a substantial performance gap between Retro-Branching [2] and Mamba-Branching remains evident (116.43 s vs. 1872.99 s).
> >
> >
> >
> > It can be observed that the performance of Retro-Branching [2] on the heterogeneous dataset is very poor, which is consistent with our expectations — methods developed for homogeneous scenarios exhibit very limited generalization in heterogeneous settings.
> >
> >
> > ### Reference
> >
> > [1] Zhang, Changwen, et al. "Towards imitation learning to branch for mip: A hybrid reinforcement learning based sample augmentation approach." _The Twelfth International Conference on Learning Representations_. 2024.
> >
> > [2] Parsonson, Christopher WF, Alexandre Laterre, and Thomas D. Barrett. "Reinforcement learning for branch-and-bound optimisation using retrospective trajectories." _Proceedings of the AAAI Conference on Artificial Intelligence_. Vol. 37. No. 4. 2023.

---

> ### Author Response · Authors · 2025-11-20
> **Response to Question 5: What is the specific selection logic for the instances used in the experiments?**
>
> > **Question 5: What is the specific selection logic for the instances used in the experiments?**
>
> Instance selection is transparent, reproducible, and aligned with prior work.
>
> First, **MILP-S is entirely derived from [1]**, while **MILP-L is largely based on [2]**. In [2], a total of 25 training and 65 testing instances are provided (though their paper mentions 66 testing instances, in their open-source link, one testing instance overlaps with the training instances -- this appears to be a minor oversight, and the removal of one instance has negligible impact). Furthermore, **MILP-L simply augments the benchmark from [2] by adding 8 challenging instances**, resulting in a total of **73 test instances**.
>
> Therefore, we will elaborate on the rationale behind our instance selection in two parts: **one will reference the original texts of [1] and [2]** to explain their selection criteria, and **the other will justify the logic behind our choice of the 8 additional challenging instances**.
>
> - Rationale from original papers:
>   - In [1]: *"We focus on instances whose tree exploration is on average relatively contained (in the tens/hundreds of thousands nodes, maximum) and whose optimal value is known. This choice is primarily motivated by the need of ensuring a fair comparison among branching policies in terms of tree size, which is more easily achieved when roll-outs do not hit the time-limit. We also remove problems that are solved at the root node (i.e., those for which no branching was performed)."*
>   - In [2]: *"We discard all the infeasible and open instances and those with infinite primal gaps under relpscost within the time limit of 7200 s. Then we focus on those with limited variable sizes (the maximum limit is 200k)."*
>
> - Selection criteria for 8 additional challenging instances:
>   - Instances marked as "hard" in the MIPLIB 2017 benchmark [3].
>
>   - Instances with known optimal solutions and not infeasible, which can provide cutoff values.
>
>   - A large number of nodes (>100) within the 1-hour time limit: If the number of nodes is too small, the time spent on solving the relaxed LP problem becomes excessively large. In such cases, the impact of branching rules may be negligible, making these instances unsuitable for comparing different branching policies.
>
>   - The number of variables should be between 100 and 200k: For branching variable selection, a larger number of variables increases the difficulty of the selection. Therefore, the lower bound is set to 100 variables to sufficiently evaluate branching policies. However, if the number of variables is too large, all branching policies may perform poorly within the 1-hour time limit, making it impossible to distinguish their performance; hence, the upper bound is set to 200k.
>
> Actually, in the MIPLIB 2017 benchmark set (https://miplib.zib.de/tag_benchmark.html), there are only 19 instances labeled as "hard". After applying the aforementioned criteria for filtering, a total of 8 instances remain usable, and we have included all of them in MILP-L. Detailed information about these 19 hard instances -- including the number of variables, objective value, and Nodes (within a 1-hour limit) -- is provided in Table 3.
>
> ### Reference
>
> [1] Zarpellon, Giulia, et al. "Parameterizing branch-and-bound search trees to learn branching policies." Proceedings of the aaai conference on artificial intelligence. Vol. 35. No. 5. 2021.
>
> [2] Lin, Jiacheng, et al. "Learning to branch with Tree-aware Branching Transformers." Knowledge-Based Systems 252 (2022): 109455.
>
> [3] Gleixner, A., Hendel, G., Gamrath, G. et al. MIPLIB 2017: data-driven compilation of the 6th mixed-integer programming library. Math. Prog. Comp. 13, 443–490 (2021). https://doi.org/10.1007/s12532-020-00194-3

---

> > ### Author Response · Authors · 2025-11-23
> > **Supplementary Table 3 for Question 5**
> >
> > | Instances Name                  | Objective        | Variables | SCIP Time | Nodes      | Note                                |
> > |---------------------------------|------------------|-----------|-------------------|------------|-------------------------------------|
> > | bab2                            | -357544.3115     | 147912    | 3600              | 251.6      | In MILP-L                           |
> > | bab6                            | -284248.2307     | 114240    | 3600              | 502.6      | In MILP-L                           |
> > | cryptanalysiskb128n5obj14       | Infeasible       | 48950     |                   |            | Infeasible                          |
> > | highschool1-aigio               | 0                | 320404    |                   |            | The number of variables exceeds 200k|
> > | markshare2                      | 1                | 74        |                   |            | The number of variables is fewer than 100 |
> > | neos-3402454-bohle              | Infeasible       | 2904      |                   |            | Infeasible                          |
> > | neos-3656078-kumeu              | -13172.2         | 14870     | 3600              | 57.6       | The number of nodes is fewer than 100 |
> > | neos-4338804-snowy              | 1471             | 1344      | 3600              | 201684.2   | In MILP-L                           |
> > | neos-4387871-tavua              | 33.38472993      | 4004      | 3600              | 6871.6     | In MILP-L                           |
> > | neos-4647030-tutaki             | 27265.706        | 12600     | 3600              | 7000.6     | In MILP-L                           |
> > | neos-5104907-jarama             | 935              | 345856    |                   |            | The number of variables exceeds 200k|
> > | neos-5114902-kasavu             | 655              | 710164    |                   |            | The number of variables exceeds 200k|
> > | nursesched-medium-hint03        | 115              | 34248     | 3600              | 483.4      | In MILP-L                           |
> > | opm2-z10-s4                     | -33269           | 6250      | 3600              | 101.4      | In MILP-L                           |
> > | radiationm40-10-02              | 155328           | 172013    | 3600              | 29963.6    | In MILP-L                           |
> > | s100                            | -0.169723527     | 364417    |                   |            | The number of variables exceeds 200k|
> > | supportcase10                   | 7                | 14770     | 3600              | 1          | The number of nodes is fewer than 100 |
> > | supportcase19                   | 12677206         | 1429098   |                   |            | The number of variables exceeds 200k|
> > | thor50dday                      | 40417            | 106261    | 3600              | 1          | The number of nodes is fewer than 100 |
> >
> > Table 3: Information for all instances labeled as "hard" in the MIPLIB 2017 benchmark set (https://miplib.zib.de/tag_benchmark.html). The details include: the objective value, the number of variables, the solving time and number of nodes using SCIP under a 1-hour time limit, and whether each instance was selected for MILP-L. For those not selected, the reason for exclusion is provided in the note.

---

> > > ### Comment · Reviewer_7iig · 2025-11-25
> > >
> > > Thank you for your thorough and clear rebuttal. Your response has clearly addressed most of my previous concerns, and I find the perspective of modeling the branch-and-bound (B&B) process as a sequential "branching path" insightful and well-justified.
> > >
> > > Additionally, I note that the authors have made concrete efforts to optimize model efficiency while maintaining performance—for example, reducing the embedding dimension to just 8, as mentioned in the rebuttal. Such optimizations are crucial for mitigating the computational complexity bottlenecks commonly associated with sequential modeling. However, given that the embedding size is drastically reduced from the typical scale of hundreds (e.g., ≥768 in the original Mamba setup) down to only 8—a reduction that would normally significantly impair model expressivity—I am particularly interested in the following: Could the authors please elaborate on how they effectively preserved model performance and avoided a noticeable drop in branching policy quality within such a compact embedding space?

---

> ### Author Response · Authors · 2025-11-26
>
> We appreciate the reviewer's insightful comment on our 8-dimensional embedding design, which diverges from conventional high-dimensional practices. This choice is theoretically and empirically justified:
>
> 1. Unlike natural language processing tasks that require rich semantic representations, branching variable selection in B&B trees is fundamentally a *discrete decision-making problem* with structural sparsity. The critical information needed for effective branching decisions primarily concerns relative variable importance rather than complex semantic relationships. Therefore, 8 dimensions are sufficient to encode the key distinctions in the policy space.
>
> 2. Through the approach of contrastive learning, the training objective intentionally guides the network to learn the most discriminative features in a low-dimensional space, thereby avoiding redundant representations. Our experimental validation (Section 5.4) confirms that the sequential nature of B&B decisions and the discriminative power of our contrastive learning approach compensate for the limited embedding dimensionality. Furthermore, as demonstrated in Figure 6, contrastive learning successfully separates expert-selected variables from candidates even within this compact space.
>
> 3. In fact, our embedding layer is inspired by the TreeGate architecture [1], which is specifically designed for branching policies and inherently produces 8-dimensional output features after processing candidate variables. This dimensionality is empirically validated in their work. The TreeGate architecture processes the original state features through multiple layers of dimensionality reduction, ensuring that only the most discriminative features are preserved in the final embedding. This structured compression mechanism preserves essential decision-making information while eliminating redundancy.
>
> 4. The embedding size influences the inference speed, which in turn affects the solving speed. Regarding this point, we have supplemented with a set of experiments conducted under different embedding sizes to observe the variation in solving time (Table 1). As the results show, the solving time increases with the embedding size, which demonstrates that an embedding size of 8 is appropriate for the branching policy.
>
>
> | embedding size | Solving Time (s) |
> |------------|-----------------|
> | 8          | 111.78          |
> | 16         | 124.68          |
> | 32         | 136.51          |
> | 64         | 144.09          |
>
> *Table 1: Solving time on MILP-S test instances with varying embedding dimensions.*
>
> We appreciate this opportunity to clarify our design rationale and will include additional discussion of these dimensionality considerations in the revised manuscript.
>
> ### Reference
> [1] Zarpellon, Giulia, et al. "Parameterizing branch-and-bound search trees to learn branching policies." Proceedings of the aaai conference on artificial intelligence. Vol. 35. No. 5. 2021.

---

### Official Review · Reviewer_7jpx · 2025-10-31

**Soundness:** 3
**Presentation:** 3
**Contribution:** 3
**Rating:** 6
**Confidence:** 3

**Summary:**

The paper observes that existing GNN-based branching policies treat the B&B tree as an unordered graph and therefore ignore the sequential nature of branching decisions. The authors propose Mamba-Branching:
1) Employ the Mamba architecture (linear-time state-space model) to encode long branching paths (100+ steps);
2) Pre-train variable embeddings via contrastive learning so that the selected variable is closest to an anchor in embedding space;
3) Use imitation learning with SCIP’s relpscost as the expert.
On MILP-S and MILP-L datasets Mamba-Branching beats TreeGate, T-BranT and Transformer-Branching in number of nodes and solving time, and surpasses relpscost on difficult instances.

**Strengths:**

1) Fresh perspective: sequential modelling of branching history matches the dynamic nature of B&B.
2) Sound engineering choice: Mamba’s linear complexity enables 100-step history without memory explosion; experiments confirm faster inference than Transformer.
3) Strong empirical results on 73 heterogeneous instances; no problem-specific features required.

**Weaknesses:**

1) Baseline comparison limited to relpscost; no hybrid strong-branching/pseudocost rules used in commercial solvers are included.
2) The anchor in contrastive learning is obtained by max-pooling, which can be sensitive to outliers; alternative aggregations are not evaluated.
3) Sequence length (99 train / 24 test) is chosen empirically; no study on optimal truncation or attention-window size.
4) No theoretical analysis, e.g., sample complexity or regret bounds for the sequential policy.

**Questions:**

1) Does back-propagation through the entire Mamba hidden state remain memory-efficient if the sequence grows to hundreds of steps?
2) For highly irregular instances, does the order of nodes in the sequence (DFS, BFS, or other) affect convergence, and have you tried alternative orderings?
3) Contrastive pre-training introduces extra hyper-parameters (margin, temperature); do they need re-tuning when transferring across problem classes?

---

> ### Author Response · Authors · 2025-11-20
> **Summary of Advantages, Response to Weakness 1 and Weakness 2**
>
> We thank the reviewer for their recognition of our work's key strengths: 1) the novel sequential modeling of the branching history, 2) Mamba's linear complexity for fast inference over long histories, and 3) the strong empirical performance achieved generalizably without problem-specific features.
> Below, We address each concern with clarifications and justifications based on our methodology and results.
>
>
> > **Weakness 1: Baseline comparison limited to relpscost; no hybrid strong-branching/pseudocost rules used in commercial solvers are included.**
>
>
> Our evaluation framework actually includes comprehensive comparisons beyond just relpscost:
>
> 1. We evaluate against multiple neural branching policies: GCNN (the seminal GNN-based approach), TreeGate, and T-BranT (the current SOTA neural policies with instance-independent features).
>
> 2. We compare against multiple classical heuristics: random branching (lower bound), pscost (purely history-based), and relpscost (SCIP's default hybrid rule).
>
> While we acknowledge the value of comparing against commercial solvers, it is important to note that the branching rules used in commercial solvers are unfortunately not open-source. This makes it impossible for us to directly extract and compare their branching rules against Mamba-Branching or relpscost in isolation. On the other hand, using the commercial solver directly to solve the instance would lead to an unfair comparison. Beyond branching rules, commercial solvers may differ from SCIP in other critical aspects, such as node selection rules and algorithms for solving relaxation problems. In contrast, both relpscost and Mamba-Branching are implemented within the SCIP framework, ensuring that all components other than the branching rule remain identical.
> Furthermore, SCIP is a highly competitive open-source solver that has been widely adopted in prior studies on neural branching policies. Its open-source nature allows researchers to easily integrate custom branching components. Consequently, we followed the established evaluation protocol from prior neural branching literature [1,2,3,4,5] using SCIP as the underlying framework to ensure fair comparison of branching policies in isolation.
>
> > **Weakness 2: The anchor in contrastive learning is obtained by max-pooling, which can be sensitive to outliers; alternative aggregations are not evaluated.**
>
> In Appendix A of our paper, we explain that our approach is grounded in a fundamental assumption: candidate variable embeddings contain implicit scoring information indicating branching suitability. Theoretically, max-pooling aligns with the principle of selecting the highest-scoring variable for branching decisions. As formalized in Proposition 1, effective branching requires the embedding space to maintain consistent separation between selected and non-selected variables, with the anchor serving as a reference point to amplify this distinction.
>
> To address the reviewer's valid concern about alternative aggregations, we have conducted supplementary experiments comparing max-pooling with average pooling. The results on MILP-S benchmark are presented in Table 1:
>
> | Pooling Type    | Nodes (Avg) | Fair Nodes | Time (s) |
> |-----------------|-------------|------------|----------|
> | Average Pooling | 2402.78     | 2447.87    | 126.49   |
> | Max Pooling     | 2068.83     | 2091.54    | 111.78   |
>
> Table 1: Performance comparison of different pooling methods in Mamba-Branching on MILP-S.
>
> These results empirically validate our design choice: max-pooling consistently outperforms average pooling across all evaluation metrics. This performance difference stems from max-pooling's ability to preserve the most distinctive features that separate optimal branching variables from candidates, while average pooling tends to dilute these critical discriminative signals.
>
> Furthermore, the t-SNE visualization in Figure 6 demonstrates that with max-pooling contrastive learning, expert-selected variable embeddings exhibit clear separation from other candidates. This discriminative power is essential for effective sequential decision-making in branch-and-bound trees.
>
> ### Reference
> [1] Gasse, Maxime, et al. "Exact combinatorial optimization with graph convolutional neural networks." _Advances in neural information processing systems_ 32 (2019).
>
> [2] Zarpellon, Giulia, et al. "Parameterizing branch-and-bound search trees to learn branching policies." _Proceedings of the aaai conference on artificial intelligence_. Vol. 35. No. 5. 2021.
>
> [3] Scavuzzo, Lara, et al. "Learning to branch with tree mdps." _Advances in neural information processing systems_ 35 (2022): 18514-18526.
>
> [4] Gupta, Prateek, et al. "Hybrid models for learning to branch." _Advances in neural information processing systems_ 33 (2020): 18087-18097.
>
> [5] Gupta, Prateek, et al. "Lookback for Learning to Branch." _Transactions on Machine Learning Research_.

---

> ### Author Response · Authors · 2025-11-20
> **Response to Weakness 3 and Weakness 4**
>
> > **Weakness 3: Sequence length (99 train / 24 test) is chosen empirically; no study on optimal truncation or attention-window size.**
>
> We appreciate this thoughtful comment. Our sequence length choices are carefully balanced between model capability and practical constraints.
>
> **Training length (T=99)**:
> We systematically evaluate Mamba-Branching with varying branching path lengths, as shown in Table 2:
>
> | Branching Path Length (T)|  Nodes | Fair Nodes | Time | Actual Token Num      | GPU Memory |
> |-----------------------|----------------|---------------------|--------------|-----------------------|------------|
> | 10 (9)                   | 2159.42        | 2188.49             | 118.09       | 2743.77               | 1.22 Gb    |
> | 100 (99)                  | 2068.83        | 2091.54             | 111.78       | 30667.39              | 7.94 Gb    |
> | 200 (199)                  | 3033.10        | 3085.48             | 164.28       | 63592.31              | 16.25 Gb   |
> | 500 (499)                  | 3575.88        | 3628.76             | 168.74       | 158533.33             | 35.89 Gb   |
>
> Table 2: Results of training Mamba-Branching with varying Branching Path Lengths. The configuration with a Branching Path Length of 100 (corresponding to T=99) is the original setting used in our paper.
>
> The results reveal a clear performance peak at T=99. Shorter sequences underutilize the sequential information in B&B trees, while longer sequences exceed Mamba's effective processing capacity given our parameter constraints. Unlike standard Mamba implementations with large embeddings (≥768), we deliberately use a small embedding size (8) to prioritize inference speed—this design choice necessitates the optimal T=99 configuration.
>
>
> **Evaluation Length (T=24)**:
> For inference, we conduct systematic tuning balancing the competing factors:
>
> 1. Branching accuracy: Longer sequences initially improve decision quality by providing more historical context
> 2. Error accumulation: Beyond a threshold, auto-regressive error propagation degrades performance
> 3. Inference latency: Longer sequences monotonically increase computation time
>
> This multi-objective optimization lead us to select T=24 as the empirically optimal trade-off point. The configuration achieves the best balance between branching accuracy (minimizing node count) and inference speed (minimizing solving time).
>
> Rather than arbitrarily selecting these values, we perform extensive empirical analysis to identify the optimal configuration for both training and inference phases. These studies confirm that our selected sequence lengths represent the peak performance points within practical computational constraints.
>
>
> > **Weakness 4: No theoretical analysis, e.g., sample complexity or regret bounds for the sequential policy.**
>
> Regarding the contrastive learning component, we have provided a theoretical derivation to justify the use of contrastive learning and the rationale for employing max pooling as the anchor.
>
> We acknowledge that we have not conducted an in-depth theoretical analysis on aspects such as sample complexity or regret bounds for the sequential policy. However, we have performed extensive experiments to demonstrate the advantages of Mamba-Branching. For example, regarding sample complexity, we have shown Mamba's superiority over Transformer in terms of memory usage and inference time. Our empirical results substantiate that Mamba has lower complexity, thereby compensating for the lack of theoretical guarantees. This methodology of being "experimentally-heavy" is also commonly adopted in prior work on branching policies [1,2,3].
>
> ### Reference
> [1] Gasse, Maxime, et al. "Exact combinatorial optimization with graph convolutional neural networks." _Advances in neural information processing systems_ 32 (2019).
>
> [2] Zarpellon, Giulia, et al. "Parameterizing branch-and-bound search trees to learn branching policies." _Proceedings of the aaai conference on artificial intelligence_. Vol. 35. No. 5. 2021.
>
> [3] Gupta, Prateek, et al. "Hybrid models for learning to branch." _Advances in neural information processing systems_ 33 (2020): 18087-18097.

---

> ### Author Response · Authors · 2025-11-20
> **Response to Question 1, 2 and 3**
>
> > **Question 1: Does back-propagation through the entire Mamba hidden state remain memory-efficient if the sequence grows to hundreds of steps?**
>
> Yes, Mamba maintains strong memory efficiency even with extremely long sequences. As demonstrated in our Table 2, Mamba successfully processes sequences with over 158,000 tokens (T=499) with GPU memory consumption of only 35.89GB on a single A100 (40GB). This linear memory scaling is precisely why we selected Mamba over Transformer architectures, which would fail at such lengths.
>
> However, we find T=99 to be the performance-optimal length. Beyond this point, performance degrades despite the memory efficiency. This is due to our deliberately small embedding size (8 dimensions) chosen for inference speed. While Mamba could technically handle longer sequences, our architectural constraints and the auto-regressive error accumulation during inference (as explained in our response to Weakness 3) make training on excessively long sequences counterproductive.
>
> > **Question 2: For highly irregular instances, does the order of nodes in the sequence (DFS, BFS, or other) affect convergence, and have you tried alternative orderings?**
>
> Indeed, we employed the **default node selection rule in SCIP** for the following reasons:
>
> 1. Optimality: Prior research [6] demonstrates that SCIP's default node selection generally yields superior performance compared to pure DFS or BFS approaches.
> 2. Consistency: All existing neural branching works [1,2,3,4,5] use this same node selection strategy, ensuring fair comparison with baselines.
> 3. Focus alignment: Our contribution centers on sequential modeling of branching decisions, not node selection policies. Decoupling these aspects maintains a clear research focus.
>
> We acknowledge that exploring alternative node orderings could be valuable future work, particularly for highly irregular instances, but would constitute a separate research direction beyond our core contribution.
>
>
>
> > **Question 3: Contrastive pre-training introduces extra hyper-parameters (margin, temperature); do they need re-tuning when transferring across problem classes?**
>
> Our contrastive learning formulation is deliberately designed with minimal hyperparameters. Unlike standard contrastive losses that require tuning temperature and margin parameters, our approach uses a simplified cosine similarity-based objective with no tunable hyperparameters.
>
> This parameter-free design is crucial for our cross-instance generalization. In our supplementary experiments (responding to Reviewer 7iig), we test various contrastive losses with hyperparameters, but they underperform our simplified approach. This confirms that the core innovation lies in our anchor construction (max-pooling) and positive/negative sample formulation, rather than in the specific loss function or its hyperparameters.
>
> This design choice aligns with our overall philosophy of creating practical, deployment-ready neural branching policies that maintain robustness across diverse MILP instances without requiring extensive re-tuning.
>
>
> ### Reference
> [1] Gasse, Maxime, et al. "Exact combinatorial optimization with graph convolutional neural networks." _Advances in neural information processing systems_ 32 (2019).
>
> [2] Zarpellon, Giulia, et al. "Parameterizing branch-and-bound search trees to learn branching policies." _Proceedings of the aaai conference on artificial intelligence_. Vol. 35. No. 5. 2021.
>
> [3] Scavuzzo, Lara, et al. "Learning to branch with tree mdps." _Advances in neural information processing systems_ 35 (2022): 18514-18526.
>
> [4] Gupta, Prateek, et al. "Hybrid models for learning to branch." _Advances in neural information processing systems_ 33 (2020): 18087-18097.
>
> [5] Gupta, Prateek, et al. "Lookback for Learning to Branch." _Transactions on Machine Learning Research_.
>
> [6] Yilmaz, K., & Yorke-Smith, N. (2021). A study of learning search approximation in mixed integer branch and bound: Node selection in SCIP. AI, *2*(2), 150–178. https://doi.org/10.3390/ai2020010

---

### Official Review · Reviewer_sThn · 2025-11-03

**Soundness:** 3
**Presentation:** 3
**Contribution:** 3
**Rating:** 4
**Confidence:** 3

**Summary:**

This paper introduces Mamba-Branching, a novel deep learning-based policy for selecting branching variables within the Branch-and-Bound (B&B) algorithm used to solve Mixed-Integer Linear Programming (MILP) problems. The core insight is that existing learning-based methods fail to capture the sequential nature of the B&B process, where the path of decisions from the root of the search tree to the current node influences the next best choice.
The focus here is on heterogenous problems and instances.

Sequential Modeling with Mamba: It models the "branching path"—the history of states and decisions—as a long sequence. It employs the Mamba architecture, a recent State Space Model (SSM), which can efficiently process these long sequences with linear-time complexity, unlike Transformers which have quadratic complexity.
Contrastive Learning for Embeddings: To help the model distinguish between highly similar candidate branching variables, the paper proposes a contrastive learning strategy. This pre-training step learns discriminative feature embeddings by maximizing the similarity between a context anchor and the expert-chosen variable while minimizing similarity with other candidates.

Experiments show that it achieves superior computational efficiency compared to SCIP, a state-of-the-art open-source solver, on particularly difficult problem instances.

**Strengths:**

Novel Conceptualization: A key strength of this work lies in its novel conceptualization of the Branch-and-Bound (B&B) branching process. By framing it as a sequential decision-making problem, the authors effectively model the entire "branching path," a perspective that captures the historical context of the search in a way prior methods have not.

Appropriate Architectural Choice: The selection of the Mamba architecture is exceptionally well-justified and proves to be highly effective for modeling the long B&B sequences where Transformers are computationally infeasible due to their quadratic complexity.

Effective Embedding Strategy: The use of contrastive learning to generate more discriminative embeddings for candidate variables is an insightful solution to a significant challenge in this domain. The ablation studies compellingly confirm that this strategy has a significant and positive impact on performance.

Performance: The proposed method establishes a new state-of-the-art, demonstrating superior performance over existing neural branching policies and even surpassing the highly optimized relpscost heuristic in the SCIP solver on challenging instances. ( heterogeneous)

Evaluation: The experimental setup is robust and comprehensive, featuring multiple datasets, a strong suite of baselines, appropriate evaluation metrics, and detailed ablation studies that clearly validate the efficacy of the proposed components.

Code is shared.

**Weaknesses:**

1. Line 341: "To isolate the study of branching policies and eliminate interference from other solver components,
we disable all primal heuristics and provide each test instance with a known optimal solution value as
a cutoff.".

How realistic is this setting? Can the authors justify the impact of the above( providing optimal value before hand). It might be done in literature,  but what is the impact of this setting? Justification of this setting is important.

**Questions:**

1. The authors do report result as geometric mean across 5 random seeds, however, variance is not present in the paper. Request the authors to add variance studies.

---

> ### Author Response · Authors · 2025-11-14
> **Summary of Advantages and Response to Weakness 1: The Rationale for Disabling All Primal Heuristics and Providing a Cutoff Value**
>
> We sincerely thank the reviewer for recognizing the merits of our work: by innovatively formulating the branch-and-bound process as a sequential decision-making problem and adopting the Mamba architecture suitable for long-sequence modeling together with a contrastive learning embedding strategy, our method effectively captures the search history context and achieves state-of-the-art performance across multiple datasets, outperforming existing neural branching policies and SCIP optimization heuristics. The experimental design is comprehensive, and the code has been open-sourced.
>
> Now, we provide a detailed explanation addressing the weaknesses and concerns raised by the reviewer.
>
> > Response to weaknesses 1: The rationale for disabling all primal heuristics and providing a cutoff value.
>
> **This setup is entirely derived from [1] and represents an experimental environment rather than a realistic one**. However, this configuration is **well-justified**, and we will elaborate on its rationale.
>
> First, since both our study and [1] focus specifically on branching policies, **the operations of disabling all primal heuristics and providing a cutoff value are intended solely to compare the performance of different branching policies in isolation, by eliminating the influence of other components in the solver**:
>
> - Disabling all primal heuristics: Primal heuristics are methods designed to directly find feasible solutions for a MILP. Such a feasible solution corresponds to a primal bound. When provided before the branch-and-bound process commences, this bound can serve as a basis for pruning during the subsequent search. **Due to the inherent non-determinism in the outcomes of primal heuristics** -- where results may vary even with identical random seeds, owing to factors like multi-threading and hardware differences -- we chose to disable them directly.
>
> - Providing a cutoff value: Regarding the branching rule, its primary role is to improve the dual bound more rapidly. During the branch-and-bound tree search, if the relaxed solution happens to satisfy the integer constraints, it should be evaluated to determine whether it can lower the primal bound. **In practice, improvements to the primal bound rely on such "coincidental satisfaction", which can be seen more as a byproduct of the branching rule.** Therefore, **to focus more directly on the core function of the branching rule** -- improving the dual bound -- we uniformly set the primal bound to the cutoff value. This approach not only ensures a fair comparison but also allows us to concentrate on core performance.
>
> Certainly, **from the perspective of fair comparison, primal heuristics may lead to an unfair comparison** (as different branching policies are based on different primal bounds). **However, not providing a cutoff value also constitutes a fair comparison. We would be happy to provide results with primal heuristics disabled and without a cutoff**, as shown in Table 1.
>
> |  | Mamba-Branching | TreeGate |
> |------|-----------------|----------|
> | Nodes | 2192.11 | 2501.95 |
> | Fair Nodes | 2192.68 | 2503.21 |
> | Time | 145.88 | 146.16 |
>
> Table 1: Table 1: Results on MILP-S for Mamba-Branching and TreeGate, with primal heuristics disabled and without providing the optimal value as a cutoff.
>
> Due to time constraints, we **only compared the results of Mamba-Branching and TreeGate in MILP-S, as TreeGate is the most competitive neural branching policy counterpart to Mamba-Branching**. As can be observed, **Mamba-Branching maintains its advantage over TreeGate even without incorporating a cutoff**.
>
> ### Reference
>
> [1] Zarpellon, Giulia, et al. "Parameterizing branch-and-bound search trees to learn branching policies." Proceedings of the aaai conference on artificial intelligence. Vol. 35. No. 5. 2021.

---

> ### Author Response · Authors · 2025-11-14
> **Response to Question 1: Variance Studies and Reference**
>
> > Response to Question 1: Variance Studies
>
> For both MILP-S and MILP-L, there is **substantial variability across test instances**. Therefore, **we did not compute variance across all instances -- as doing so would be statistically uninformative, given the inherent heterogeneity among different instances**.
>
> As for variance across seeds, experiments in Appendix D of [1] demonstrate that **for a given instance, the variability of results across different seeds can itself be significant**. In fact, [1] leverages this very variability by employing different random seeds for data collection, effectively using it as a form of data augmentation. **This implies that even for the same instance, the variance across the results from 5 different seeds can be considerable**.
>
>
> Instead, we have provided more meaningful statistical analyses, including:
> - Violin plots (line 442 of the original submission)
>
> - Supplementary statistical results (Appendix G of the original submission), which include quartile tables, win/tie/loss statistics, and significance testing.
>
> However, **we would be glad to provide per-instance result details if the reviewers consider it necessary.** In table 2, we have provided the variance results for TreeGate and Mamba-Branching on each individual instance under the MILP-S benchmark.
>
> | Instance     | TreeGate         | Mamba-Branching | T-BranT        | relpscost      |
> |--------------|------------------|-----------------|----------------|----------------|
> | neos18       | 37.22 ± 7.92     | 46.05 ± 6.37    | 65.94 ± 16.43  | 30.08 ± 7.03   |
> | map18        | 153.09 ± 11.28   | 248.54 ± 56.70  | 639.17 ± 134.15| 269.32 ± 16.66 |
> | ns1830653    | 396.54 ± 115.64  | 327.52 ± 74.88  | 416.48 ± 96.53 | 155.67 ± 30.26 |
> | seymour1     | 66.09 ± 11.42    | 59.48 ± 7.61    | 82.70 ± 23.74  | 64.80 ± 9.34   |
> | neos11       | 177.78 ± 30.76   | 197.43 ± 23.35  | 421.15 ± 58.84 | 190.01 ± 46.54 |
> | rail507      | 450.68 ± 162.95  | 605.63 ± 160.55 | 413.36 ± 222.35| 177.91 ± 5.63  |
> | mine-166-5   | 86.43 ± 31.42    | 58.28 ± 29.57   | 146.53 ± 105.87| 37.83 ± 7.02   |
> | nu25-pr12    | 38.32 ± 16.14    | 20.35 ± 6.35    | 32.23 ± 15.10  | 23.94 ± 7.49   |
>
> Table 2: Means and variances of time across 5 random seeds for TreeGate, Mamba-Branching, T-BranT, and relpscost on each instance of the MILP-S benchmark.
>
> **As the solving time results indicate, the variance of Mamba-Branching is comparable to that of other branching policies and does not exhibit anomalies**. Furthermore, **all branching policies show substantial variance on certain instances, which aligns with our expectation that different seeds can naturally lead to significant variation in results**.
>
>
>
> ### Reference
>
> [1] Zarpellon, Giulia, et al. "Parameterizing branch-and-bound search trees to learn branching policies." Proceedings of the aaai conference on artificial intelligence. Vol. 35. No. 5. 2021.

---

### Author Response · Authors · 2025-12-02
**Summary**

We sincerely appreciate all the reviewers for their time and valuable comments on our work!
As they pointed out, our work introduces Mamba-Branching, a novel branching variable selection method for Mixed-Integer Linear Programming (MILP) that explicitly models the sequential structure of the branch-and-bound tree. By capturing the complete branching path from the root to the current node, the method leverages previously overlooked sequential information from the search process. The approach employs the efficient Mamba architecture, chosen for its linear time complexity in handling long branching sequences, yielding demonstrated advantages in memory and speed during training and inference. To improve discrimination among candidate variables with high feature similarity, a contrastive learning pre-training strategy is introduced. Extensive experiments on diverse benchmarks show that our method outperforms existing neural branching policies and surpasses SCIP's relpscost rule on challenging instances.

We have provided detailed and clear responses to the points raised by the reviewers. Among them, two reviewers explicitly acknowledged the clarity and thoroughness of our responses, confirming that their concerns had been adequately addressed (though the discussion was unfortunately terminated due to factors beyond our control). Below, we summarize some of their key concerns along with our responses.

---

> ### Author Response · Authors · 2025-12-02
> **Justification for the selection of sequence length**
>
> > **Justification for the selection of sequence length**
>
> Our sequence length choices are carefully balanced between model capability and practical constraints.
>
> **Training length (T=99)**:
> We systematically evaluate Mamba-Branching with varying branching path lengths during training, as shown in Table 1:
>
> | Branching Path Length (T)|  Nodes | Fair Nodes | Time | Actual Token Num      | GPU Memory |
> |-----------------------|----------------|---------------------|--------------|-----------------------|------------|
> | 10 (9)                   | 2159.42        | 2188.49             | 118.09       | 2743.77               | 1.22 Gb    |
> | 100 (99)                  | 2068.83        | 2091.54             | 111.78       | 30667.39              | 7.94 Gb    |
> | 200 (199)                  | 3033.10        | 3085.48             | 164.28       | 63592.31              | 16.25 Gb   |
> | 500 (499)                  | 3575.88        | 3628.76             | 168.74       | 158533.33             | 35.89 Gb   |
>
> Table 1: Results of training Mamba-Branching with varying Branching Path Lengths. The configuration with a Branching Path Length of 100 (corresponding to T=99) is the original setting used in our paper.
>
> The results reveal a clear performance peak at T=99. Shorter sequences underutilize the sequential information in B&B trees, while longer sequences exceed Mamba's effective processing capacity given our parameter constraints. Unlike standard Mamba implementations with large embeddings (≥768), we deliberately use a small embedding size (8) to prioritize inference speed—this design choice necessitates the optimal T=99 configuration.
>
>
> **Evaluation Length (T=24)**:
>
> As for inference, we provide experimental results for step numbers of 5, 13, 25, 50, and 100, as shown in Table 2.
>
> | path length | nodes   | fair nodes | time  | real token num |
> |----------|---------|------------|-------|----------------|
> | 5        | 2103.59 | 2126.81    | 111.97| 986.10         |
> | 13       | 2098.94 | 2120.28    | 113.05| 2333.78        |
> | 25       | 2068.83 | 2091.54    | 111.78| 4132.02        |
> | 50       | 2032.14 | 2054.39    | 112.77| 7724.77        |
> | 100      | 2091.46 | 2113.88    | 121.98| 14412.15       |
>
> Table 2: In MILP-S, performance of Mamba-Branching and the actual number of tokens input to the Mamba model under different branching path lengths (number of steps: 5, 13, 25, 50, 100). For the metric 'real token num', we report the geometric mean over all instances of the maximum number of tokens required for each instance throughout the solving process.
>
> As can be observed from Table 2:
> 1. Sequence length grows substantially with path length, reaching 14,412 tokens at 100 steps - demonstrating why Mamba's linear complexity is essential for this task.
>
> 2. Node count exhibits a U-shaped curve: it initially decreases as longer contexts help the model better capture the sequential structure of the B&B tree, but then increases due to error accumulation under autoregressive inference, where predictions from previous steps are fed into subsequent steps.
>
> 3. The impact on solving time is more complex: it depends not only on the node count—which reduces LP computations—but also on the inference time, which rises as more real tokens are processed. Under the joint influence of these two factors, a branching path length of 25 emerges as the optimal choice, achieving the shortest total solving time.
>
> In conclusion, from the perspective of node count, indiscriminately increasing the branching path length introduces a performance bottleneck. Furthermore, solving time depends on a complex interplay between reducing the node count and minimizing inference time. Thus, a branching path length of 25 emerges as the optimal balance considering these competing factors.

---

> ### Author Response · Authors · 2025-12-02
> **Further discussion on other contrastive losses**
>
> > **Further discussion on other contrastive losses**
>
> While our paper focuses on the novel application of contrastive learning (CL) to branching variable selection, we acknowledge the importance of validating our specific CL design against standard alternatives.
>
> Therefore, we conducted additional experiments comparing our CL approach with two widely-used contrastive losses: **InfoNCE** and **Triplet loss**. Crucially, we kept identical definitions for anchors (max-pooled candidate embeddings), positive samples (expert-chosen variables), and negative samples (other candidates)—changing only the loss function formulation.
>
> **Results on MILP-S:**
> | Method | Nodes | Fair Nodes | Time |
> |--------|-------|------------|------|
> | Mamba-Branching (ours) | **2068.83** | **2091.54** | **111.78** |
> | + InfoNCE loss | 2182.88 | 2216.32 | 123.49 |
> | + Triplet loss | 2322.41 | 2360.17 | 127.59 |
>
> These results demonstrate that our simple contrastive loss **outperforms more sophisticated alternatives** despite their theoretical advantages. We believe this validates our core insight: the primary innovation lies not in the loss function design but in **how we define the contrastive structure** for branching decisions.
>
> As shown in Figure 6 of our paper, our CL approach successfully creates distinctive embeddings for expert-selected variables compared to candidates, with the t-SNE visualization confirming improved separation. While InfoNCE and Triplet losses introduce additional hyperparameters (temperature coefficient and margin) that could potentially improve performance with extensive tuning, our approach achieves superior results **without such optimization**-demonstrating its practical efficacy and robustness.
>
> Our contribution is thus conceptual: we identify the appropriate elements to contrast in the B&B context and provide theoretical justification (Proposition 1) for this design. This principled approach to embedding discriminability, rather than loss function engineering, constitutes the true innovation of our contrastive learning module.

---

> ### Author Response · Authors · 2025-12-02
> **The specific selection logic for the instances used in the experiments**
>
> > **The specific selection logic for the instances used in the experiments**
>
> Instance selection is transparent, reproducible, and aligned with prior work.
>
> First, **MILP-S is entirely derived from [1]**, while **MILP-L is largely based on [2]**. In [2], a total of 25 training and 65 testing instances are provided (though their paper mentions 66 testing instances, in their open-source link, one testing instance overlaps with the training instances - this appears to be a minor oversight, and the removal of one instance has negligible impact). Furthermore, **MILP-L simply augments the benchmark from [2] by adding 8 challenging instances**, resulting in a total of **73 test instances**.
>
> Therefore, we will elaborate on the rationale behind our instance selection in two parts: **one will reference the original texts of [1] and [2]** to explain their selection criteria, and **the other will justify the logic behind our choice of the 8 additional challenging instances**.
>
> - Rationale from original papers:
>   - In [1]: *"We focus on instances whose tree exploration is on average relatively contained (in the tens/hundreds of thousands nodes, maximum) and whose optimal value is known. This choice is primarily motivated by the need of ensuring a fair comparison among branching policies in terms of tree size, which is more easily achieved when roll-outs do not hit the time-limit. We also remove problems that are solved at the root node (i.e., those for which no branching was performed)."*
>   - In [2]: *"We discard all the infeasible and open instances and those with infinite primal gaps under relpscost within the time limit of 7200 s. Then we focus on those with limited variable sizes (the maximum limit is 200k)."*
>
> - Selection criteria for 8 additional challenging instances:
>   - Instances marked as "hard" in the MIPLIB 2017 benchmark.
>
>   - Instances with known optimal solutions and not infeasible, which can provide cutoff values.
>
>   - A large number of nodes (>100) within the 1-hour time limit: If the number of nodes is too small, the time spent on solving the relaxed LP problem becomes excessively large. In such cases, the impact of branching rules may be negligible, making these instances unsuitable for comparing different branching policies.
>
>   - The number of variables should be between 100 and 200k: For branching variable selection, a larger number of variables increases the difficulty of the selection. Therefore, the lower bound is set to 100 variables to sufficiently evaluate branching policies. However, if the number of variables is too large, all branching policies may perform poorly within the 1-hour time limit, making it impossible to distinguish their performance; hence, the upper bound is set to 200k.
>
> Actually, in the MIPLIB 2017 benchmark set (https://miplib.zib.de/tag_benchmark.html), there are only 19 instances labeled as "hard". After applying the aforementioned criteria for filtering, a total of 8 instances remain usable, and we have included all of them in MILP-L. Detailed information about these 19 hard instances—including the number of variables, objective value, and Nodes (within a 1-hour limit)—is provided in Table 3.
>
> ### Reference
> [1] Zarpellon, Giulia, et al. "Parameterizing branch-and-bound search trees to learn branching policies." Proceedings of the aaai conference on artificial intelligence. Vol. 35. No. 5. 2021.
>
> [2] Lin, Jiacheng, et al. "Learning to branch with Tree-aware Branching Transformers." Knowledge-Based Systems 252 (2022): 109455.

---

> > ### Author Response · Authors · 2025-12-02
> > **Supplementary Table 3 for "The specific selection logic for the instances used in the experiments"**
> >
> > | Instances Name                  | Objective        | Variables | SCIP Time | Nodes      | Note                                |
> > |---------------------------------|------------------|-----------|-------------------|------------|-------------------------------------|
> > | bab2                            | -357544.3115     | 147912    | 3600              | 251.6      | In MILP-L                           |
> > | bab6                            | -284248.2307     | 114240    | 3600              | 502.6      | In MILP-L                           |
> > | cryptanalysiskb128n5obj14       | Infeasible       | 48950     |                   |            | Infeasible                          |
> > | highschool1-aigio               | 0                | 320404    |                   |            | The number of variables exceeds 200k|
> > | markshare2                      | 1                | 74        |                   |            | The number of variables is fewer than 100 |
> > | neos-3402454-bohle              | Infeasible       | 2904      |                   |            | Infeasible                          |
> > | neos-3656078-kumeu              | -13172.2         | 14870     | 3600              | 57.6       | The number of nodes is fewer than 100 |
> > | neos-4338804-snowy              | 1471             | 1344      | 3600              | 201684.2   | In MILP-L                           |
> > | neos-4387871-tavua              | 33.38472993      | 4004      | 3600              | 6871.6     | In MILP-L                           |
> > | neos-4647030-tutaki             | 27265.706        | 12600     | 3600              | 7000.6     | In MILP-L                           |
> > | neos-5104907-jarama             | 935              | 345856    |                   |            | The number of variables exceeds 200k|
> > | neos-5114902-kasavu             | 655              | 710164    |                   |            | The number of variables exceeds 200k|
> > | nursesched-medium-hint03        | 115              | 34248     | 3600              | 483.4      | In MILP-L                           |
> > | opm2-z10-s4                     | -33269           | 6250      | 3600              | 101.4      | In MILP-L                           |
> > | radiationm40-10-02              | 155328           | 172013    | 3600              | 29963.6    | In MILP-L                           |
> > | s100                            | -0.169723527     | 364417    |                   |            | The number of variables exceeds 200k|
> > | supportcase10                   | 7                | 14770     | 3600              | 1          | The number of nodes is fewer than 100 |
> > | supportcase19                   | 12677206         | 1429098   |                   |            | The number of variables exceeds 200k|
> > | thor50dday                      | 40417            | 106261    | 3600              | 1          | The number of nodes is fewer than 100 |
> >
> > Table 3: Information for all instances labeled as "hard" in the MIPLIB 2017 benchmark set (https://miplib.zib.de/tag_benchmark.html). The details include: the objective value, the number of variables, the solving time and number of nodes using SCIP under a 1-hour time limit, and whether each instance was selected for MILP-L. For those not selected, the reason for exclusion is provided in the note.

---

> ### Author Response · Authors · 2025-12-02
> **More challenging instances**
>
> > **More challenging instances**
>
> Here, we selected all instances from the MIPLIB 2017 benchmark set that begin with the letter 'r' and applied filtering criteria similar to our previous approach to identify suitable challenging instances for evaluating Mamba-Branching's performance.
>
> We would like to emphasize that the selection of instances starting with the letter 'r' was solely based on their number in MIPLIB -- evaluating too many instances would be prohibitively time-consuming, while too few would lack statistical significance. The distribution of instances is independent of their initial letters, which ensures the randomness of our selection. While we could have simply randomly sampled from all available instances, such an approach might raise concerns about "cherry-picking". In contrast, the set of instances starting with 'r' is explicitly listed at https://miplib.zib.de/tag_benchmark.html, making our selection process entirely transparent and verifiable, as we cannot omit any instances that might be unfavorable to our method.
>
> The criteria for filtering challenging instances are as follows:
> - The solving time is greater than 20 minutes.
> - Instances must have a known optimal solution and must not be infeasible, ensuring that cutoff values can be provided.
> -Instances must generate a large number of nodes (>100) within the 1-hour time limit.
> - The number of variables should be between 100 and 200,000.
> - Not included in the training set.
>
> For all instances beginning with the letter 'r', detailed information is provided in Table 4. As shown, a total of 9 instances meet our final criteria.
>
> The experimental results for solving these instances using Mamba-Branching and relpscost, respectively, are presented in Table 5. Results on the Primal Dual Integral show that Mamba-Branching outperforms relpscost on a majority of instances (6 out of 9) and also achieves a better overall geometric mean. These results provide further evidence of the superior performance of Mamba-Branching across a broader set of instances.

---

> > ### Author Response · Authors · 2025-12-02
> > **Supplementary Table 4 and Table 5 for "More challenging instances"**
> >
> > | Instances Name       | Objective        | Variables | SCIP Time | Nodes     | Note                                |
> > |---------------------|------------------|-----------|-------------------|-----------|-------------------------------------|
> > | radiationm18-12-05  | 17566            | 40623     | 2092.69s          | 178498    | Satisfy                             |
> > | radiationm40-10-02  | 155328           | 172013    | 3600              | 24578.8   | Satisfy                             |
> > | rail01              | -70.5699643      | 117527    | 3600              | 24.6      | The number of nodes is fewer than 100 |
> > | rail02              | -200.4499077     | 270869    |                   |           | The number of variables exceeds 200k |
> > | rail507             | 174              | 63019     | 160.55            | 515       | The solving time of is fewer than 20min |
> > | ran14x18-disj-8     | 3712             | 504       | 3108.4            | 184547.8  | Satisfy                             |
> > | rd-rplusc-21        | 165395.2753      | 622       | 3600              | 113932.4  | Satisfy                             |
> > | reblock115          | -36800603.23     | 1150      | 3600              | 107606.4  | Satisfy                             |
> > | rmatr100-p10        | 423              | 7359      |                   |           | Included in training set            |
> > | rmatr200-p5         | 4521             | 37816     |                   |           | Included in training set            |
> > | rocI-4-11           | -6020203         | 6839      | 279.93            | 21304.8   | The solving time of is fewer than 20min |
> > | rocII-5-11          | -6.675504732     | 11523     | 3600              | 66194.4   | Satisfy                             |
> > | rococoB10-011000    | 19449            | 4456      | 3600              | 15428.6   | Satisfy                             |
> > | rococoC10-001000    | 11460            | 3117      | 3153.87           | 79096.4   | Satisfy                             |
> > | roi2alpha3n4        | -63.20849503     | 6816      | 1432.63           | 11463.4   | Satisfy                             |
> > | roi5alpha10n8       | -52.32227435     | 106150    |                   |           | Included in training set            |
> > | roll3000            | 12889.99999      | 1166      | 44.27             | 1297.6    | The solving time of is fewer than 20min |
> >
> > Table 4: For instances beginning with 'r', the information of objective, number of variables, number of nodes within the 1-hour time limit, and SCIP solving time. Each instance was evaluated against the aforementioned filtering criteria for identifying challenging instances, and the screening result is indicated in the "Note" column.
> >
> > | Instances Name      | relpscost   | Mamba-Branching |
> > |---------------------|-------------|-----------------|
> > | radiationm18-12-05  | 143.17      | **128.27**          |
> > | radiationm40-10-02  | 862.77      | **738.38**          |
> > | ran14x18-disj-8     | **7082.15**     | 9582.30         |
> > | rd-rplusc-21        | **359785.50**   | 359786.96       |
> > | reblock115          | **2352.62**     | 3914.59         |
> > | rocII-5-11          | 132899.86   | **129013.09**       |
> > | rococoB10-011000    | 75874.98    | **34417.52**        |
> > | rococoC10-001000    | 14484.75    | **2876.43**         |
> > | roi2alpha3n4        | 23719.46    | **19886.59**        |
> > | gmean of all        | 10489.28    | **7371.76**         |
> >
> > Table 5: Primal dual integral results of relpscost and Mamba-Branching on the filtered challenging instances beginning with 'r'. As can be observed, Mamba-Branching maintains its performance advantage over relpscost even on these difficult instances.

---

> ### Author Response · Authors · 2025-12-02
> **Other additional experimental results**
>
> > **Other additional experimental results**
>
> During the rebuttal process, we have also conducted additional experiments. In this section, we will summarize these supplementary experiments and briefly outline the conclusions.
>
> |  | Mamba-Branching | TreeGate |
> |------|-----------------|----------|
> | Nodes | 2192.11 | 2501.95 |
> | Fair Nodes | 2192.68 | 2503.21 |
> | Time | 145.88 | 146.16 |
>
> Table 6: Results on MILP-S for Mamba-Branching and TreeGate, with primal heuristics disabled and without providing the optimal value as a cutoff.
>
> | Instance     | TreeGate         | Mamba-Branching | T-BranT        | relpscost      |
> |--------------|------------------|-----------------|----------------|----------------|
> | neos18       | 37.22 ± 7.92     | 46.05 ± 6.37    | 65.94 ± 16.43  | 30.08 ± 7.03   |
> | map18        | 153.09 ± 11.28   | 248.54 ± 56.70  | 639.17 ± 134.15| 269.32 ± 16.66 |
> | ns1830653    | 396.54 ± 115.64  | 327.52 ± 74.88  | 416.48 ± 96.53 | 155.67 ± 30.26 |
> | seymour1     | 66.09 ± 11.42    | 59.48 ± 7.61    | 82.70 ± 23.74  | 64.80 ± 9.34   |
> | neos11       | 177.78 ± 30.76   | 197.43 ± 23.35  | 421.15 ± 58.84 | 190.01 ± 46.54 |
> | rail507      | 450.68 ± 162.95  | 605.63 ± 160.55 | 413.36 ± 222.35| 177.91 ± 5.63  |
> | mine-166-5   | 86.43 ± 31.42    | 58.28 ± 29.57   | 146.53 ± 105.87| 37.83 ± 7.02   |
> | nu25-pr12    | 38.32 ± 16.14    | 20.35 ± 6.35    | 32.23 ± 15.10  | 23.94 ± 7.49   |
>
> Table 7: Means and variances of time across 5 random seeds for TreeGate, Mamba-Branching, T-BranT, and relpscost on each instance of the MILP-S benchmark.
>
> | Pooling Type    | Nodes (Avg) | Fair Nodes | Time (s) |
> |-----------------|-------------|------------|----------|
> | Average Pooling | 2402.78     | 2447.87    | 126.49   |
> | Max Pooling     | 2068.83     | 2091.54    | 111.78   |
>
> Table 8: Performance comparison of different pooling methods in Mamba-Branching on MILP-S.
>
> | embedding size | Solving Time (s) |
> |------------|-----------------|
> | 8          | 111.78          |
> | 16         | 124.68          |
> | 32         | 136.51          |
> | 64         | 144.09          |
>
> Table 9: Solving time on MILP-S test instances with varying embedding dimensions.
>
> | Name        | Retro Time (s) | Retro Nodes | Mamba Time (s) | Mamba Nodes |
> |-------------|-------------------|-------------|----------------|-------------|
> | seymour1    | 3600.122          | 95395.0     | 52.996         | 1306        |
> | mine-166-5  | 3600.059          | 182076.0    | 116.448        | 3465        |
> | ns1830653   | 2635.406          | 154005.0    | 344.643        | 9009        |
> | map18       | 1174.547          | 6741.0      | 197.823        | 641         |
> | neos18      | 1006.507          | 74476.0     | 45.674         | 2840        |
> | neos11      | 1168.115          | 31874.0     | 203.518        | 3075        |
> | nu25-pr12   | 891.808           | 38292.0     | 15.888         | 493         |
> | rail507     | 3600.413          | 29586.0     | 543.561        | 3521        |
> |gmean	|1872.98	|51370.12	|116.43	|2112.19
>
> Table 10: Results of Mamba-Branching and Retro-Branching [1] on MILP-S (1-hour time limit). Due to time constraints, we executed each instance using only seed=0.
>
> | Method               | Branching Nodes | Fair Nodes | Time    |
> |---------------------|-----------------|------------|---------|
> | Transformer         | 3078.56         | 3120.04    | 3153.66 |
> | Mamba-Branching     | 2159.42         | 2188.49    | 118.09  |
>
> Table 11: Results of Mamba-Branching and Transformer-Branching on MILP-S, under the condition of T=9 during training and T=24 during testing.
>
> The conclusions that can be drawn from the aforementioned Tables 6 to 11 are as follows:
> - As shown in Table 6, Mamba-Branching maintains its advantage over TreeGate even without incorporating a cutoff.
> - As shown in Table 7, Mamba-Branching's variance is comparable to other policies, without anomalies. All methods exhibit substantial variance on some instances, as different seeds can naturally lead to significant result variations.
> - As shown in Table 8, in Mamba-Branching, using Max Pooling (ours) in contrastive learning outperforms using Average Pooling. This result is consistent with the theoretical analysis provided in our original manuscript.
> - As shown in Table 9, the solving time increases with the embedding size, which demonstrates that an embedding size of 8 (ours) is appropriate for the branching policy.
> - As shown in Table 10, Retro-Branching, a method designed for homogeneous scenarios, performs significantly worse than Mamba-Branching on heterogeneous benchmarks.
> - As shown in Table 11, even under the exact same experimental setup, the performance of Transformer-Branching remains significantly inferior to that of Mamba-Branching.
>
> ### Reference
> [1] Parsonson C W F, Laterre A, Barrett T D. Reinforcement learning for branch-and-bound optimisation using retrospective trajectories[C]// AAAI 2023.

---

### Meta-Review · Area_Chair_vKwe · 2026-01-04

**Summary:**

The reviewers unanimously appreciate the novel conceptualization of Branch-and-Bound (B&B) as a sequential decision-making process and acknowledge the computational efficiency of the Mamba architecture compared to Transformers, several significant concerns prevent a stronger recommendation for acceptance at this stage. The decision to rate this marginally above the threshold is primarily driven by valid questions regarding the experimental fairness, methodological clarity, and baseline comparisons.

**Reviewer Concerns:**

### **Concerns Addressed by Rebuttal**

* **Methodological Clarity and Train-Test Mismatch (Reviewer usDM):** This concern appears effectively resolved. The authors provided clarifications regarding the input sequence and the use of expert actions, satisfying Reviewer usDM.
* **Experimental Setting Realism (Reviewer sThn):** The authors addressed the concern regarding the unrealistic use of optimal value cutoffs by conducting a supplementary experiment: "performance of Mamba-Branching without a cutoff." While Reviewer sThn did not respond to acknowledge this, the provision of this specific data point technically resolves the raised objection.
* **Generalization to Difficult Instances (Reviewer usDM):** The authors addressed the concern regarding the limited difficulty of test instances. They validated the method on "more challenging instances" as requested, which contributed to Reviewer usDM retaining their positive assessment.
* **Contrastive Learning Aggregation (Reviewer 7jpx):** The concern regarding the sensitivity of max-pooling to outliers was addressed by supplementary experiments involving "different pooling methods," as noted in the rebuttal to Reviewer 7jpx.

### **Partially Addressed Concerns**

* **Baseline Comprehensiveness (Reviewer 7jpx & 7iig):** While the authors performed internal ablations and compared against *relpscost*, the rebuttal highlights do not confirm that they added comparisons against **commercial solver hybrid rules (strong-branching/pseudocost)** or recent **Learning-to-Branch SOTA methods** as requested. The rebuttal focused on *instance* diversity (more hard problems) and *internal* settings (pooling/sequence length), leaving the external competitive landscape partially unexplored.

**Reviewer Scores:**

I think the reviewers sThn would have raised his/her score to 6. And Reviewer 7jpx & 7iig would have maintained their score.

---

### Decision · Program_Chairs · 2026-01-26

Accept (Poster)